# Image-based machine learning models for customized soil moisture management

Yooan Kim[1], Taehyeong Kim[2,3], Sungyong Lee[4], Suhyun Lee[5], Kyo Suh[2,5,6,7]*

**1** Institute of Construction and Environmental Engineering, Seoul National University, Seoul, South Korea, **2** Integrated Major in Global Smart Farm, Seoul National University, Seoul, South Korea, **3** Department of Biosystems Engineering, Seoul National University, Seoul, South Korea, **4** Samsung SDS, Seoul, South Korea, **5** Graduate School of International Agricultural Technology, Seoul National University, Pyeongchang, South Korea, **6** Institute of Green Bio Science Technology, Seoul National University, Pyeongchang, South Korea, **7** Artificial Intelligence Institute, Seoul National University, Seoul, South Korea

* kyosuh@snu.ac.kr

## Abstract

Crop growth can vary even under the same cultivation conditions, highlighting the limitations of conventional smart farming systems that apply uniform treatments to all crops. These average-based approaches often overlook individual plant needs shaped by microenvironments and physiological differences, resulting in inefficient resource use and reduced yields. While crop-specific management is important for improving productivity, there is a lack of non-invasive methods to monitor soil conditions at the individual plant level. This study presents an AI-based system that combines soil sensors and image analysis to support customized moisture management. Transplanted wild-simulated ginseng was used as a model crop. RGB images of the soil surface were collected with sensor data from different depths (3 cm, 10 cm, and 15 cm) to capture vertical moisture distribution. Several deep learning models were evaluated for predicting surface moisture, with DenseNet121 showing the highest accuracy ($R^2$ = 97.3%, RMSE = 4.14). For deeper soil layers, the random forest regression model achieved the best performance ($R^2$ = 90.6%, RMSE = 4.97), effectively capturing nonlinear moisture dynamics. These results demonstrate that surface image data can be used to estimate soil moisture non-invasively and enable data-driven, plant-specific crop management systems. This research provides a foundation for data-driven, customized, soil moisture management in smart farming. Future studies should focus on validating the model across diverse crops and soil types, and integrate additional spectral data to enhance its robustness and scalability.

## 1. Introduction

Climate-smart agriculture is a technological innovation that transforms the paradigm of crop cultivation by minimizing the impact of natural environments on agriculture [1].

**Data availability statement:** All relevant data are within the manuscript.

**Funding:** This research was supported by the Artificial Intelligence Institute, Seoul National University, and the Youlchon Foundation (Nongshim Corporation and its affiliated companies) through the Youlchon AI Scholarship Grant in 2023 (to Y.K.). This research was supported by the Industry–Academia–Research Collaboration Activation Support (R&D) Program (Project No. 2025-25422608 to K.S.) funded by the Ministry of Science and ICT (MSIT), Republic of Korea.

**Competing interests:** The authors have declared that no competing interests exist.

It simultaneously addresses climate change while enhancing agricultural productivity and minimizing environmental impacts through efficient resource utilization [2–5]. Smart agriculture, including precision agriculture and smart farms, plays a pivotal role in achieving the goals of climate-smart agriculture [6–10]. Smart farms focusing on controlled environment agriculture initially emerged as first-generation remote management systems. These systems provided monitoring and control functions for agricultural environments [11–13]. However, they were limited to simple environmental control. Subsequently, smart farms evolved into second-generation intelligent systems that autonomously controlled environments, used accumulated data, and adapt flexibly to climate change [14,15]. The next generation of smart farms is projected to evolve beyond simple environmental control by incorporating sophisticated systems capable of customized management tailored to the growth status of individual crops [16,17]. Smart farming technology is a critical tool for minimizing resource waste and maximizing productivity in complex agricultural environments affected by climate change [18,19]. It is also essential foundational technology for realizing climate-smart agriculture [1,20].

In the evolution of smart farming, water resource management technology has gained significant attention [21,22]. As climate change intensifies water scarcity, efficient water resource management in agriculture has become an increasingly critical challenge [23]. Moisture management in crop cultivation is a key factor in determining productivity and quality [24]. For example, inadequate moisture management can cause direct economic losses by reducing plant growth, increasing pest and disease occurrence, and decreased yields [23]. Precise moisture management requires accurately understanding crop water requirements based on growth stages and environmental conditions, and then providing the appropriate amount of water for each plant at the right time [24,25]. Since each soil layer plays a distinct role in crop growth, understanding soil moisture variation by depth is also essential [26]. The upper layer reflects rapid irrigation and evaporation changes [27], the middle layer is the primary zone for root activity, and the bottom layer functions as a reservoir that buffers excess moisture [26,28].

Despite the critical importance of precise moisture management, most current smart farm systems employ uniform management approaches that fail to consider individual crop characteristics. These systems have primarily focused on optimizing environmental factors such as temperature, humidity, and lighting [29,30]. However, plants grown under identical conditions vary in their physiological traits and growth rates, making standardized approaches insufficient [31]. With climate change and resource scarcity intensifying, maximizing productivity while using resources efficiently has become a necessity [23]. Customized management optimizes each plant's growth while minimizing resource waste [32,33]. Thus, it serves as a key strategy for agricultural productivity and sustainability [1,20]. Smart farm systems incorporating AI models can analyze complex agricultural data to determine optimized conditions for individual plants. Such models can potentially overcome conventional limitations [34]. Implementing such customized management requires innovative technological solutions that accurately monitor individual plant conditions and translate the data into actionable management strategies.

Most artificial intelligence research related to smart farms incorporating various digital technologies has primarily focused on uniform management at the field or greenhouse level [33,35,36]. Limited studies exist on customized cultivation management systems that consider the growth status of individual crops [37–39]. Previous studies have mainly emphasized developing models that manage overall soil conditions or facility environments based on information from large-scale monitoring systems using satellite data or drones [40–42]. However, these models have limitations in reflecting the detailed growth differences of individual crops. A remote-management, IoT-based smart irrigation system has been developed to support soil-moisture management at both field and greenhouse scales, demonstrating meaningful gains in water-resource conservation and crop productivity [43]. Beyond systems for managing moisture that affects crop growing environments, various studies have conducted moisture monitoring using remote sensing. Satellite-based remote sensing has also been applied to improve watering management at both regional and farm scales. In addition, small-sized satellite systems offering high spatial and temporal resolution have been utilized to assess crop water use in relatively narrow areas [44] Their approach mitigated limitations of wide coverage and low resolution inherent in conventional satellite data. However, despite the reduction in target scale, limitations remain. Detailed management is necessary at the individual plant level to implement customized cultivation management based on the growth status of individual crops. For customized cultivation to become more widespread, simpler methods are required beyond installing sensors on each plant. The purpose of this study is to evaluate and compare the performance of various image-based machine learning algorithms for customized management based on the growth status of individual crops using image data associated with sensor data. Transplanted wild-simulated ginseng, a moisture-sensitive and high-value crop, was selected as the target plant. RGB images of the soil surface were collected during the early germination stage, along with sensor-based soil moisture data from surface, middle, and deep layers. Multiple deep learning models, including ResNet and EfficientNet, were evaluated for surface-level prediction. Additionally, this study applied regression models by soil depth for moisture estimation, including polynomial regression, support vector regression, and random forest regression. This approach enables non-destructive, image-based estimation of soil moisture, facilitating individualized crop management strategies tailored to the specific environmental and physiological conditions of each plant.

## 2. Materials and methods

### 2.1. Customized growth management setup

#### 2.1.1. Target crop: Wild-simulated Ginseng.
High value-added agricultural and forest products command premium prices relative to their yield, highlighting the need for customized management to maximize productivity. The market price per gram of wild-simulated ginseng (WSG) exceeds that of cultivated ginseng by more than 20-fold, highlighting its prominence as a high-value forest product [45,46]. Most WSG is marketed after more than seven years of cultivation. The quality and price are determined by various factors such as region, shape, weight, and other attributes that affect market value [47–49]. Since the survival rate of WSG decreases dramatically from the fourth year onward, effective management strategies are needed to improve survival rates and produce high-quality ginseng [50,51]. To establish legal standards for WSG and clearly distinguish it from cultivated ginseng, we investigated changes in WSG characteristics after a short post-treatment period following pre-germination transplantation. Accordingly, five-year-old WSG cultivated in Pyeongchang, Gangwon Province, was selected and transplanted into experimental soil as the target forest product for this study.

#### 2.1.2. Experiment setup.
Changes in major saponin components were tracked to develop a standardized cultivation technique. This post-treatment technology confirmed that the composition and weight of WSG (three years or older) could increase to levels comparable to six-year-old ginseng within a short period of 90 days (maximum 150 days) (SNU R&DB Foundation, 2023 [52]). Image data collection associated with sensor data for soil moisture management was conducted while performing WSG cultivation using methods that are identical to the patented post-treatment technology. The WSG specimens used in the experiment had an average weight of 0.954g, with diverse weights ranging from 0.401g to 2.488g to ensure experimental representativeness. The transplantation of WSG was performed in indoor facilities, with each

specimen placed in an individual pot. The transplanted WSG underwent post-treatment in the Genesis Farmbot system (Fig 1).

WSG seedlings were grown under controlled greenhouse conditions following transplantation. The image-based analysis in this study focused on the early growth period after transplantation, a stage during which wild-simulated ginseng is highly sensitive to moisture conditions. Moisture stress during this period has a direct influence on seedling survival and early physiological responses, including changes in saponin composition [53]. To ensure consistent rooting conditions across plants, we used a standardized peat moss and perlite substrate in a 6:4 ratio, which is commonly adopted for ginseng seedling cultivation [54,55]. This substrate composition is known to provide high porosity, low bulk density, and a favorable balance between aeration and water-holding capacity. Such physical characteristics create a uniform and supportive rooting environment, which is particularly important during the moisture-sensitive early growth stage of WSG.

Following transplantation, the soil moisture content was adjusted to $50 \pm 10\%$ of saturation, based on two-point calibration (oven-dry and fully saturated conditions), to provide a suitable starting environment. In this study, soil moisture content refers to the relative saturation of the soil, expressed as a percentage with 0% for oven-dry soil and 100% for saturation. Hereafter, this is referred to simply as soil moisture content. Fig 2 presents representative images of the soil surface used in the experiment. The presence of mixed materials such as peat moss and perlite made it difficult to visually distinguish differences in soil moisture levels. All other environmental conditions, except for moisture, remained consistent across groups. This experiment aimed to quantitatively assess the effects of early-stage moisture stress on WSG growth after transplantation and to identify optimal moisture management strategies for this critical stage. All experiments were conducted within institutional facilities, and no specific permits were required for this research.

## 2.2. Soil moisture sensor system

The core hardware components of the soil moisture sensing system included an Arduino Mega 2560, an ESP8266 WiFi module, and capacitive soil moisture sensors (Table 1, Fig 3). We managed all sensor data for customized soil moisture management of WSG through an integrated data acquisition system. We developed data collection and transmission code using the Arduino IDE, enabling each sensor to transmit data to a cloud server (ThingSpeak) via WiFi. This system allows for real-time or interval-based of soil moisture conditions. By continuously monitoring moisture levels, the system supports optimized moisture management tailored to the specific needs of the crop.

Soil moisture status was measured using capacitive moisture sensors integrated into the Arduino-based moisture management system. Each sensor was installed individually for each WSG plant to enable customized monitoring and

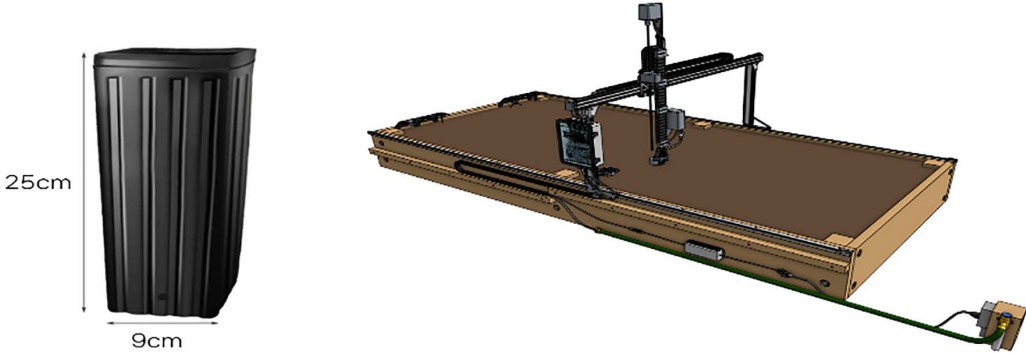

**Fig 1. Experimental setup for wild-simulated ginseng cultivation.** The left image shows the pot specifications used for cultivation, measuring 25 cm in height and 9 cm in diameter, which provides an optimal environment for root development. The right image illustrates the cultivation system of using a Genesis FarmBot (bed size: 1.5 m × 3 m, height: 0.5 m), equipped with an X-Y-Z gantry.

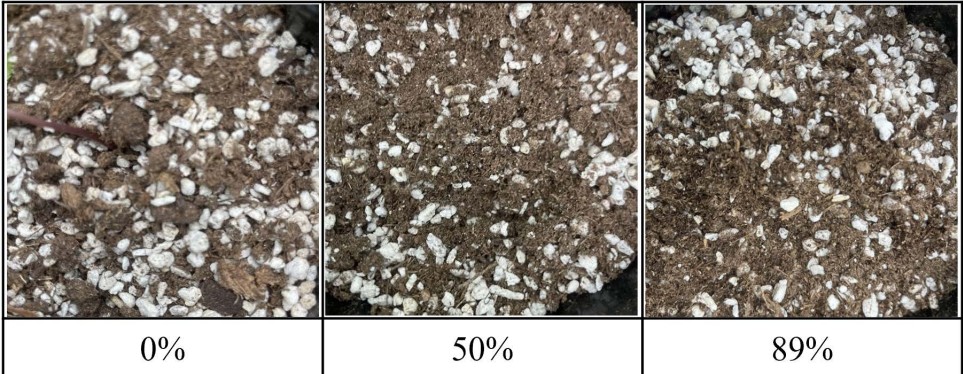

| 0% | 50% | 89% |

**Fig 2. Sample soil surface images at different moisture levels.** These images are provided solely for qualitative illustration, and no spatial or physical scale is implied (not to scale).

**Table 1. Hardware and Software Development Environment.**

| Type | Hardware/Software | Function and role |
|---|---|---|
| Hardware | Arduino 2560 | Arduino Board |
| | Capacitive Soil Moisture Sensor | Moisture sensor for measuring water content in the soil |
| | ESP8266 & adapter | Microcontroller with built-in WiFi capabilities |
| Software | Arduino IDE | Software platform used for writing, compiling, and uploading code to Arduino boards |
| | ThingSpeak | Cloud server enabling data collection and storage for IoT applications |

control of growth conditions. To avoid signal distortion, which is commonly caused by corrosion in resistive sensors, we used HW-390 capacitive soil moisture sensors. The capacitive soil moisture sensors used in this study detect changes in dielectric permittivity associated with varying soil moisture levels based on frequency domain reflectometry (FDR). These readings are converted into electrical signals ranging from 0 to 1023. Higher values correspond to lower moisture conditions. To convert the raw sensor outputs into normalized moisture (%) values, we applied a two-point calibration procedure using oven-dry (0%) and fully saturated (100%) reference states. The average sensor readings were $542 \pm 54$ at 0% and $344 \pm 50$ at 100%. In this study, the moisture (%) values represent a normalized index derived from the calibrated sensor signals rather than direct volumetric or gravimetric water content.

For the soil-depth analysis experiment, we installed three moisture sensors per plant at distinct heights above the pot base: 15 cm (top layer), 10 cm (middle layer), and 3 cm (deep layer). These layer designations were adopted to align with conventional soil profile terminology and to provide clearer interpretation of the soil structure within the pot.

### 2.3. Image-based prediction system

**2.3.1. Image collection.** We collected image data associated with the sensor system using a readily available RGB camera (iPhone 12 Pro), simulating practical, real-world conditions. To minimize variations in image brightness caused by sunlight we captured all images between 1:00 PM and 3:00 PM. The image acquisition was conducted under natural lighting conditions within the greenhouse, rather than in a controlled lighting setup, to evaluate the feasibility of predicting soil moisture levels using AI in actual cultivation environments. As outlined in Table 2 and Fig 4, images were continuously captured from immediately after transplantation until the soil surface became partially obscured by plant foliage during early growth.

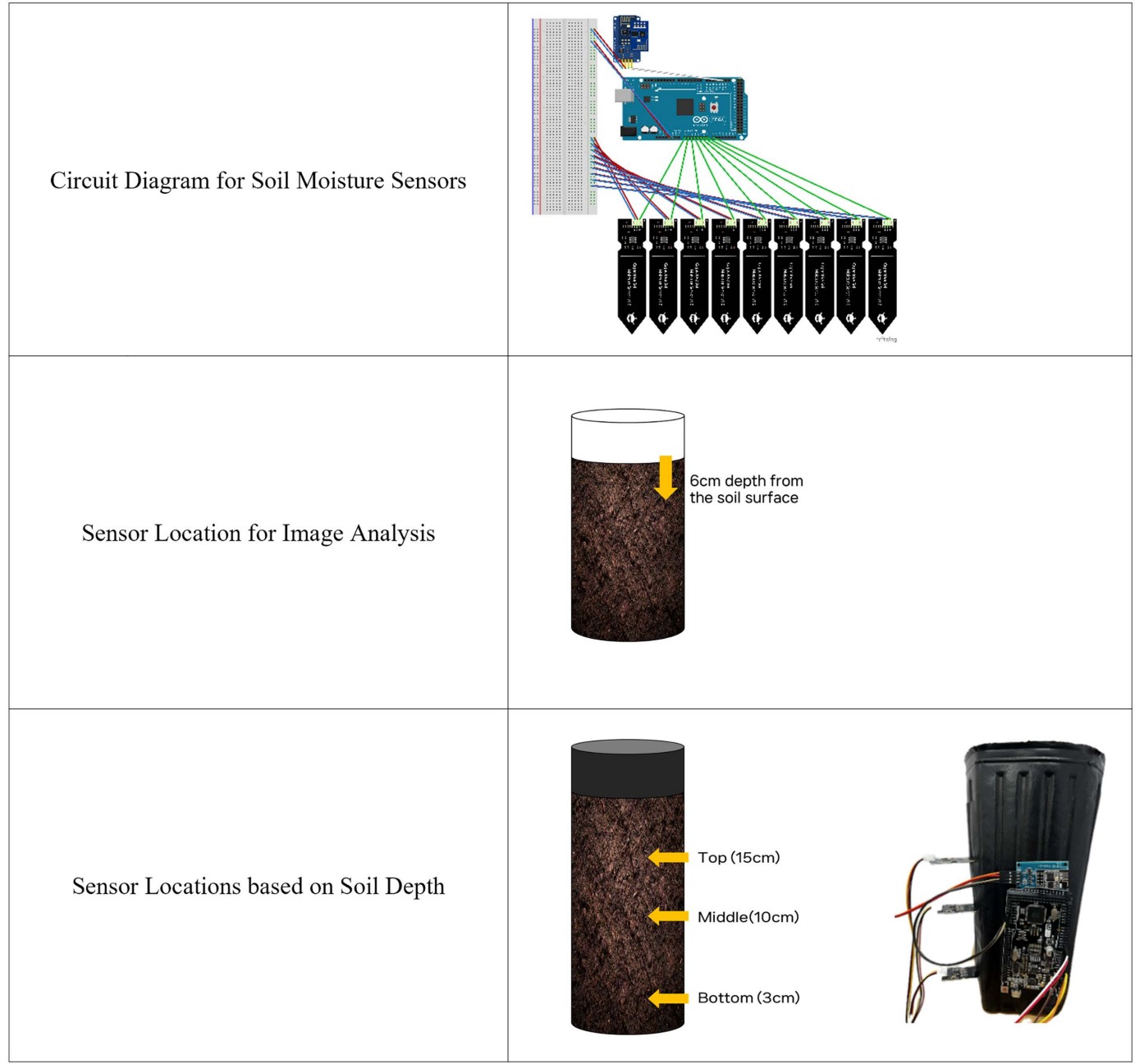

**Fig 3. Soil moisture sensing system and sensor placement.**

**2.3.2. Image preprocessing.** A hybrid sampling strategy combining data augmentation and oversampling was applied to mitigate class imbalance and improve the generalization performance of the model. The target number of samples was set based on the moisture level with the highest sample count. For moisture levels with fewer samples, we performed oversampling with replacement until the target was reached. We then applied data augmentation techniques, including rotation, translation, scaling, and flipping, to further diversify the training data. This preprocessing ensured that all moisture

**Table 2. Camera Setting for in situ image acquisition.**

| Focal Length | 26 mm |
| --- | --- |
| Aperture | F1.6 |
| ISO Sensitivity | Auto |
| Camera Orientation | Portrait |
| Shooting Equipment | Camera bracket utilized |
| Focusing Distance | 200mm |
| Digital Image Dimensions | 4032 x 3024 pixels |
| Camera Resolution | 12MP |

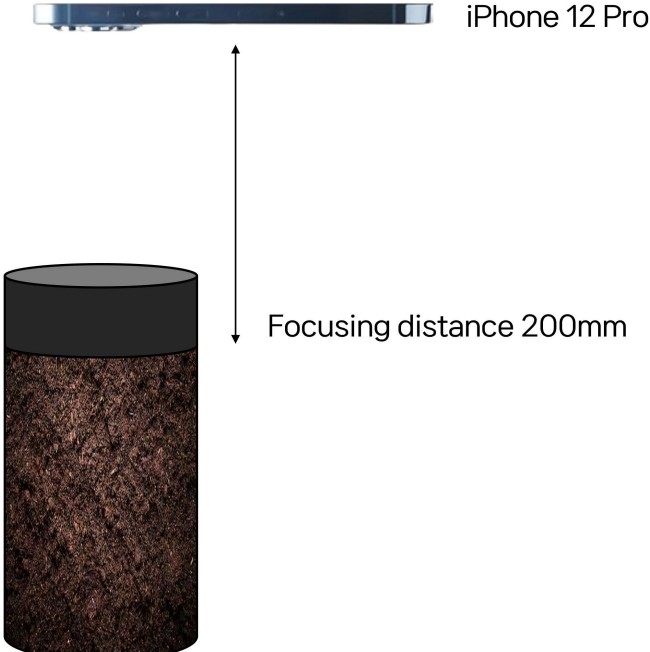

iPhone 12 Pro

Focusing distance 200mm

**Fig 4. Camera Setting for in situ image acquisition.**

levels were represented with an equal number of samples, preventing the model from overfitting to any specific range. As a result, the model was able to learn more diverse visual patterns across different soil moisture conditions.

We processed images using OpenCV's IMREAD function to convert raw pixel data into RGB matrices and resized each image to 224 × 224 pixels (Fig 5). To maintain model accuracy, we normalized pixel values to a range between 0 and 1, ensuring consistent input values across the dataset [56,57]. The RGB color space was preserved throughout preprocessing without conversion to grayscale or HSV. We chose this approach because the deep learning models used in this study (i.e., DenseNet121 and InceptionV3) were pre-trained on RGB images and optimized to extract informative features from color channels. Therefore, preserving the RGB format allowed the model to leverage its full capacity for feature extraction and contributed to stable performance without additional color space transformation.

**2.3.3. Customized soil moisture prediction model.** To develop a deep neural network (DNN) regression model for customized soil moisture management based on soil images, we used Python with the TensorFlow Keras framework. We adopted six pre-trained deep learning architectures—ResNet50 (He et al., 2016), EfficientNetB0 (Tan & Le, 2019),

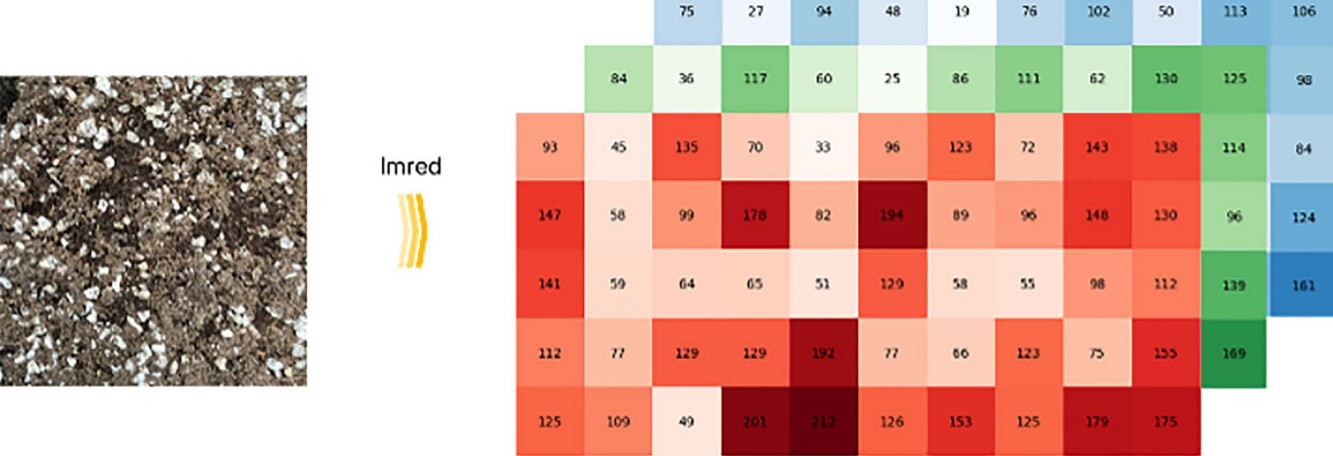

**Fig 5. Input and processing of soil images.** The original soil image on the left is converted into a three-channel RGB matrix. This process uses IMRED to decompose pixel values into red, green, and blue channels. The resulting matrix is input data for feature learning in deep learning models.

MobileNetV2 (Sandler et al., 2018), InceptionV3 (Szegedy et al., 2016), DenseNet121 (Huang et al., 2017), and NASNetMobile (Zoph et al., 2018). All were originally trained on the ImageNet dataset. Although these architectures are primarily designed for image classification, we adapted them for regression by modifying their final layers to output continuous values. We employed transfer learning, removing the original classification layers and adding fully connected layers to support regression, enabling prediction of continuous soil moisture values from images.

The model architecture consisted of four main blocks, as summarized in Table 3. A pre-trained convolutional neural network (Base Model) received input images with a shape of 224 × 224 × 3 and used weights trained on the ImageNet dataset. The output was a 7 × 7 × C feature map, where C denotes the number of feature channels, varying by architecture. To reduce parameters and prevent overfitting, we applied global average pooling to compress the 3D feature map into a 1D feature vector of length C. This process was followed by a dense layer with 1,024 neurons and a ReLU activation function to capture high-level representations. The final output layer consisted of a single neuron with a linear activation function for continuous soil moisture prediction. The number of trainable parameters varied by architecture, which affected both the training time and performance. Hyperparameters were selected through a trial-and-error process [58–60]. The final configurations were optimized for regression performance. We selected these architectures for their suitability in soil

**Table 3. (Top)** Comparison of pre-trained CNN architectures, including input/output shapes, number of features, and total parameters. All models take 224 × 224 × 3 input images and generate architecture-specific feature maps. **(Bottom)** Architectural overview of the soil moisture prediction model, consisting of a pre-trained CNN base followed by modified top layers.

| | Input Shape | Output of Last Convolution Layer | Number of Features | Total Parameters |
|---|---|---|---|---|
| DenseNet121 | 224 × 224 × 3 | 7 x 7 x 1024 | 1024 | 8.0M |
| EfficientNetB0 | | 7 x 7 x 1280 | 1280 | 5.3M |
| InceptionV3 | | 7 x 7 x 2048 | 2048 | 23.9M |
| MobileNet | | 7 x 7 x 1024 | 1024 | 4.2M |
| NASNetMobile | | 7 x 7 x 1056 | 1056 | 5.3M |
| ResNet50 | | 7 x 7 x 2048 | 2048 | 25.6M |

 

moisture prediction. They are categorized into three types based on their design characteristics: (1) standard deep CNNs, (2) complex, high-capacity extractors, and (3) lightweight, efficiency-focused models.

ResNet50, a deep and complex architecture, employs skip connections that enable stable training of deep networks and facilitate the extraction of complex patterns. Despite its effectiveness, it involves a relatively large number of parameters. DenseNet121 incorporates dense connectivity, which encourages feature reuse, alleviates the vanishing gradient problem, and improves memory efficiency through parameter sharing [61]. EfficientNetB0 achieves strong performance with fewer parameters by applying compound scaling, making it suitable for extracting soil image features efficiently. MobileNetV2, designed for lightweight applications, offers fast inference speed. However, its performance may slightly degrade when dealing with highly complex patterns [61,62]. InceptionV3 captures multi-scale features simultaneously, which is advantageous for identifying patterns of varying sizes. This benefit, however, comes at the cost of increased structural complexity. NASNetMobile, optimized through a neural architecture search, provides automatically designed models. Nevertheless, it may exhibit training instability, particularly on small datasets [63,64].

### 2.4. Depth-based soil moisture monitoring system

**2.4.1. Data collection and preprocessing.** This study constructed an additional estimation model to predict subsurface soil moisture using top-view images. We added this model because measuring only the surface moisture is insufficient to capture the actual content in deeper layers. A depth-specific soil moisture prediction approach was also adopted to estimate the lower-layer moisture conditions. To eliminate the influence of plant uptake, we conducted the experiment using pots without planted crops. We applied water from the top using a standard irrigation method, which ensured consistent control of the watering process and accurate tracking of moisture changes at each depth. We installed soil moisture sensors at three heights above the pot base. The sensors were positioned 15 cm, 10 cm, and 3 cm above the base, corresponding to the top, middle, and deep soil layers, respectively. This configuration allowed for accurate collection of soil moisture data at multiple depths.

We performed data preprocessing to improve the model performance and enhance the accuracy of the predictions. We removed data points where all sensors recorded maximum values (100%) immediately after irrigation, because these oversaturated conditions were not informative. Next, we filtered out error-prone data showing unrealistic distributions, such as when the upper or middle sensor reported higher moisture levels than the bottom sensor. We also excluded cases where the upper layer recorded higher moisture than the middle layer. Finally, we eliminated outliers using the inter-quartile range (IQR) method. Values outside the Q1–Q3 range were removed to increase reliability and integrity. The initial dataset consisted of 888 samples. After preprocessing, 519 valid samples remained and were used for model training and evaluation.

**2.4.2. Soil moisture prediction model by depth.** We designed the model to predict deep-layer moisture based on readings from the top and middle layers. The input variables were the soil moisture measurements obtained from the top and middle layers (15 cm and 10 cm above the pot base), and the target variable was defined as the soil moisture measured in the deep layer at 3 cm above the base. This configuration enabled the estimation of lower-layer moisture levels from surface and mid-layer conditions.

Several nonlinear regression models were evaluated to predict soil moisture variations across different soil depths. The soil-water characteristic curve (SWCC), which describes the relationship between soil matric potential and moisture content, typically follows a nonlinear pattern. This characteristic served as the basis for applying nonlinear regression methods to improve the accuracy of predictions. To identify the most effective model, we compared six nonlinear regression approaches: polynomial regression, support vector regression (SVR), bagging decision trees, random forest regression, k-nearest neighbors (KNN) regression, and gradient boosting. These models were selected for their ability to capture complex, nonlinear patterns. By applying and comparing these models on depth-based soil moisture data, the goal was to select the optimal model that best reflects the physical properties and hydrodynamic behavior of the soil.

Polynomial regression incorporates polynomial terms of the independent variable to model higher-order nonlinear relationships. This approach was used to capture the complex moisture distribution patterns observed across different soil depths. SVR models nonlinear relationships between soil moisture and depth in a high-dimensional space using kernel functions. Because of its robustness against outliers, SVR is particularly suitable for handling uncertainties in field-measured data [65,66]. Bagging decision trees and random forest regression, which are both tree-based ensemble models, offer the advantage of capturing complex interactions among the various factors influencing soil moisture dynamics. In particular, random forest models estimate nonlinear relationships by aggregating multiple decision trees, enhancing prediction stability [67,68]. KNN regression was applied based on the assumption that nearby measurement points share similar soil physical properties, such as porosity and particle size distribution [69,70]. Moisture values recorded at spatially close locations are likely to influence predictions. Thus, gradient boosting was selected for its ability to incrementally learn complex nonlinear variations in soil moisture. This model also allows for the evaluation of the relative importance of different predictive factors.

We optimized hyperparameters for the nonlinear regression models by balancing model complexity and generalization performance, following Occam's razor. This principle suggests that among competing hypotheses that explain the same phenomenon, the simplest one is preferable. In the context of machine learning, Occam's razor serves as a guideline for selecting models that are neither overly complex nor prone to overfitting. Evaluating predictive performance requires not only accuracy on training data but also the ability to generalize to unseen data. To assess generalization, the cross-validated R² score was used as the primary evaluation metric. A model was considered well-generalized when the difference between its training R² and cross-validation R² remained within 10% [71,72]. Each nonlinear regression model was tuned using its optimal set of hyperparameters and then applied to the soil moisture prediction task. The final model was selected based on comparative performance, with the goal of identifying the approach that best captured the physical characteristics and hydrodynamic behavior of the soil.

## 3. Experiments

### 3.1. Model training and experimental setup

We implemented the deep learning models using TensorFlow (Keras) on a Windows platform equipped with an Intel i9-11900K CPU, NVIDIA RTX 3050 GPU (Santa Clara, CA, USA), and 32GB of RAM.

We divided the image dataset into training, validation, and test sets in a 6:2:2 ratio. This allocation was based on theoretical justifications from prior studies. To ensure reliable evaluation, at least 20% of the total data was reserved for testing, which meets the minimum requirement for assessing statistical significance at the 95% confidence level [73]. When applying the hold-out validation approach, allocating 20–30% of the data for validation has been shown to be effective for assessing generalization performance and detecting overfitting [74]. According to Vapnik-Chervonenkis (VC) theory, assigning 60% of the data to the training set provides a sufficient sample size for stable learning [75]. This split ratio addressed the bias-variance trade-off by ensuring adequate sizes for validation and test sets, reducing variance in performance estimates, and providing stable evaluation metrics [76].

During the training, we tested three learning rates (0.00001, 0.0001, and 0.001) and up to 100 epochs to compare model performance. Model optimization was guided by standard regression metrics, which were monitored during training to evaluate model convergence and stability. We treated the number of epochs as a critical parameter for balancing performance and efficiency. When no meaningful improvement appeared, we reduced the number of epochs to avoid unnecessary computation. To prevent overfitting, the optimal number of epochs for each model was determined based on the point at which performance stabilized during early training. All deep learning models were trained using the mean squared error (MSE) loss function and the Adam optimizer.

We also split the depth-based dataset into training, validation, and test sets in a 6:2:2 ratio. To improve model stability and convergence speed, all soil moisture values were normalized to a range between 0 and 1 using a MinMax scaler. A

batch size of 32 was used in the data loader configuration. Nonlinear regression models for predicting moisture content at each soil depth were evaluated under various hyperparameter settings. For random forest regression, the number of trees varied from 20 to 100. SVR was tested with different regularization values (C) ranging from 0.1 to 1000. For KNN regression, we adjusted the number of neighbors from 3 to 11 to identify the optimal configuration.

## 3.2. Performance metrics

Evaluating regression model performance was essential to assess how well the models fit the data. In this study, MSE was selected as the loss function for training deep neural network (DNN) regression models. MSE penalizes large errors more heavily, encouraging the model to focus on reducing them (Goodfellow et al., 2016). We used standard regression metrics to evaluate model performance, including MSE, MAE, RMSE, and R². These metrics assess the average error, sensitivity to outliers, and explained variance, respectively. The formulas are shown in Equation (1) through (4) MAE calculates the average of absolute differences between predicted and actual values, providing a direct measure of typical error size. RMSE, computed as the square root of MSE, estimates the average prediction error with greater sensitivity to large deviations. R² measures the proportion of variance in the observed data explained by the model and is widely used for predictive accuracy. We applied these evaluation metrics to assess both accuracy and suitability of the models, and to select the optimal predictive approach for soil moisture estimation.

$$MAE = \frac{1}{n} \sum_{i=1}^{n} |y_i - \widehat{y_i}|$$

(1)

$$MSE = \frac{1}{n} \sum_{i=1}^{n} (y_i - \widehat{y_i})^2$$

(2)

$$RMSE = \sqrt{\frac{1}{n} \sum_{i=1}^{n} (y_i - \widehat{y_i})^2}$$

(3)

$$R^2 = 1 - \frac{\sum_{i=1}^{n} (y_i - \widehat{y_i})^2}{\sum_{i=1}^{n} (y_i - \overline{y_i})^2}$$

(4)

where y represents the measured value, $\hat{y}$ is the predicted value by the models, $\bar{y}$ is the average value of the measured value, and n is the number of samples.

## 4. Results

### 4.1. Performance of image-based prediction system

**4.1.1. Comparison of deep learning models.** We compared six deep learning architectures (ResNet50, EfficientNetB0, MobileNetV2, InceptionV3, DenseNet121, and NASNetMobile) based on their performance metrics and model complexity. As shown in Table 4, the upper part summarizes each model's R², RMSE, learning rate and the number of training epochs. Fig 6 provides a graphical comparison illustrating the number of parameters, MAE, and R² for each architecture. Models positioned in the lower-left quadrant of the graph demonstrate superior performance, characterized by fewer parameters, lower computational cost, and reduced MAE. The size of each bubble indicates the difference between the validation MAE and test MAE, with smaller bubbles reflecting stronger generalization performance.

Among the evaluated models, DenseNet121 demonstrated the best overall performance for soil moisture prediction, achieving the lowest MAE (2.07) and highest validation R² (97.3%), despite using only 8.08 million parameters. Its dense

**Table 4. Performance comparison of deep learning architectures for soil moisture prediction. The upper section summarizes R², RMSE, learning rate, and epoch for six models. DenseNet121 showed the highest prediction accuracy, while NASNetMobile performed the poorest.**

|  | R² | RMSE | LR | Epoch |
|---|---|---|---|---|
| DenseNet121 | 97.3 | 4.14 | 0.00001 | 68 |
| EfficientNetB0 | 96.9 | 4.99 | 0.001 | 69 |
| InceptionV3 | 96.9 | 4.20 | 0.001 | 69 |
| ResNet50 | 96.6 | 8.81 | 0.0001 | 52 |
| MobileNetV2 | 95.8 | 5.20 | 0.001 | 79 |
| NASNetMobile | 89.7 | 8.39 | 0.00001 | 100 |

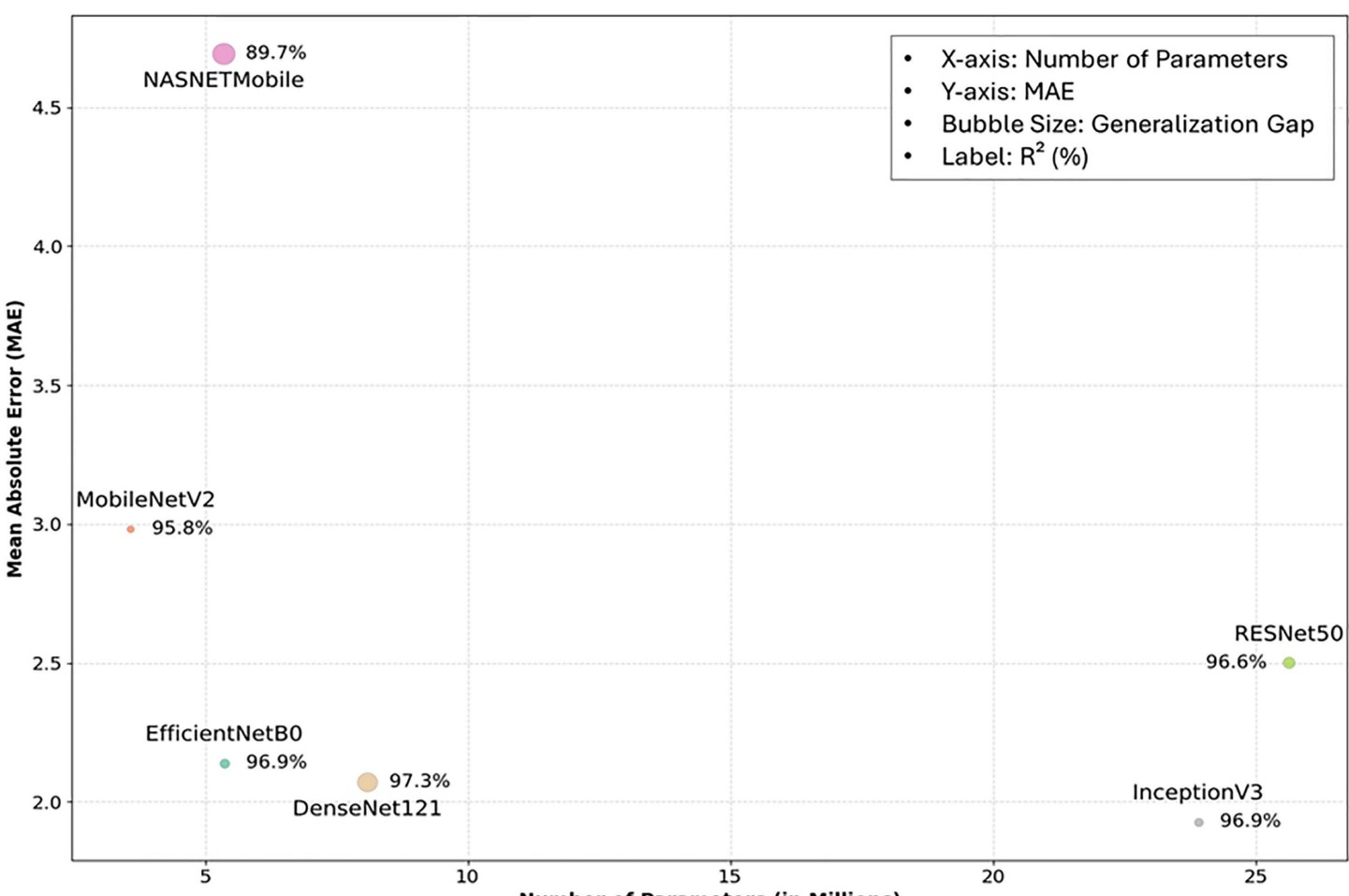

**Fig 6. Comparison of model complexity, MAE, and generalization gap.** Bubble size denotes generalization gap and labels indicate R². DenseNet121 achieved the best trade-off between accuracy and efficiency, while NASNetMobile showed poor generalization despite low complexity.

connectivity structure, which enables direct feature reuse across layers, enhanced information flow and preserved spatial resolution, allowing the model to effectively capture fine-grained texture variations in soil images. This architectural efficiency led to minimal prediction errors, making DenseNet121 the optimal model for practical image-based soil moisture

management. EfficientNetB0 also performed strongly, with an MAE of 2.14 and validation $R^2$ of 96.9%, using only 5.36 million parameters. Its configuration (learning rate: 0.001, 69 epochs) achieved a strong balance between model complexity and predictive power, highlighting its suitability for lightweight applications.

ResNet50 and InceptionV3 required significantly more computational resources, approximately 25.6 million and 23.9 million parameters, respectively, which is about 4.5 to 4.8 times greater than EfficientNetB0. Despite the increased complexity, both models demonstrated comparable performance in soil moisture prediction. ResNet50 was optimized with a learning rate of 0.0001 and 52 training epochs. Its skip connection structure helped mitigate the vanishing gradient problem in deep layers, allowing the model to effectively distinguish distinct texture regions within the soil images. InceptionV3, trained with a learning rate of 0.001 over 69 epochs, employed convolutional filters of varying sizes to capture the soil structure at multiple spatial scales. Both models achieved high predictive performance, with $R^2$ values of 96.6% for ResNet50 and 96.9% for InceptionV3. The small differences between validation MAE and test MAE indicated strong generalization capabilities. Although InceptionV3 incurred higher computational cost, its accuracy and reliability suggest that it remains a promising candidate for image-based soil moisture prediction.

NASNetMobile had a parameter count similar to EfficientNetB0 (5.35 million), but it yielded the poorest performance among the six architectures, with an MAE exceeding 4.5 and an $R^2$ of 89.7%. Even though it used a low learning rate of 0.00001 and trained for 100 epochs, the model underperformed. This limitation likely stemmed from the nature of neural architecture search (NAS), which can cause overfitting to specific patterns while overlooking broader feature generalization. MobileNetV2, with 3.57 million parameters, features a lightweight architecture and achieved an $R^2$ of 95.8% and an MAE of 2.98 when trained with a learning rate of 0.001 for 79 epochs. The model effectively captured textures and patterns across global regions of the soil images. Notably, MobileNetV2 demonstrated the smallest gap between validation and test MAE, indicating superior generalization compared to the other architectures. However, although NASNetMobile and MobileNetV2 are designed for resource-constrained environments and they exhibit low computational demand, their performance in soil moisture prediction was relatively limited.

**4.1.2. Actual vs. predicted in deep learning models.** The predictive performance of each deep learning architecture was further evaluated by comparing the actual and predicted soil moisture values. DenseNet121 and InceptionV3 demonstrated the highest levels of agreement, showing minimal prediction error across most data points. DenseNet121 achieved the most accurate results by leveraging its dense connectivity structure for efficient feature reuse, while InceptionV3 benefited from multi-scale convolutional filters that captured diverse spatial features. ResNet50 and EfficientNetB0 also achieved strong predictive accuracy. Their architectural strengths (e.g., skip connections in ResNet50 and compound scaling in EfficientNetB0) contributed to improved texture recognition and consistent moisture estimation. These trends are visually illustrated in Fig 7, which presents scatter plots of actual versus predicted values for each model.

## 4.2. Depth-wise analysis

**4.2.1. Soil moisture characteristics by depth.** This study analyzed the variation in soil moisture content at different soil depths. As expected, deeper soil layers exhibited higher moisture levels, primarily due to reduced exposure to evaporation. Table 5 presents the descriptive statistics for soil moisture measured at three depths: deep, middle, and top. A total of 519 samples were collected for each depth, including the mean, standard deviation, minimum, maximum, and quartiles. The deep layer showed the highest mean soil moisture content, measured as relative saturation, with a mean value of 65.33% and a standard deviation of 15.01%. In contrast, the middle layer recorded a significantly lower mean of 41.00%, with a standard deviation of 11.74%. The top layer exhibited the lowest mean moisture content at 21.26% and the lowest variability (SD = 8.29%), indicating more stable moisture conditions near the surface. These results reveal a clear stratification of moisture levels by depth, with a notable decrease from deep to top. This pattern is attributed to water movement following irrigation, where moisture gradually percolates downward, accumulating more in the lower layers. The

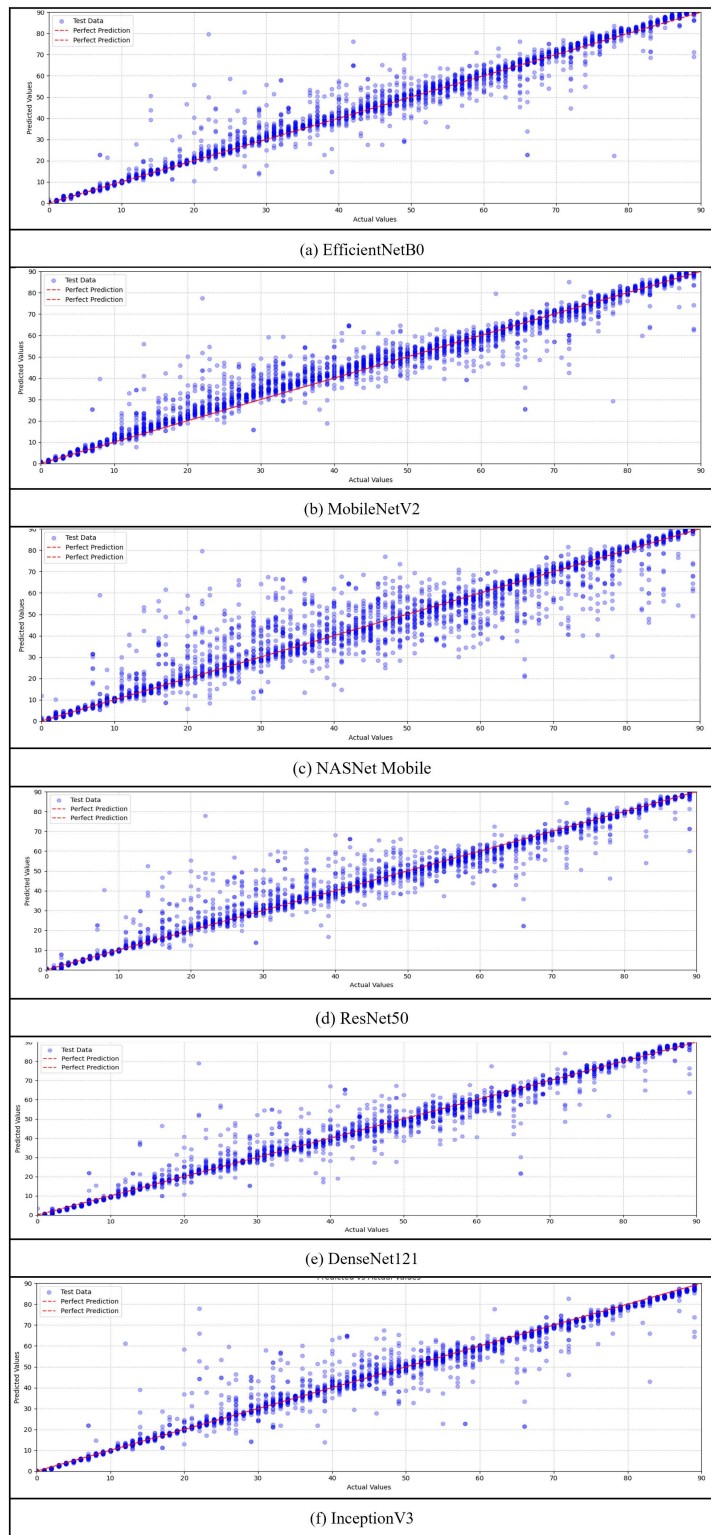

**Fig 7. Comparison of actual and predicted values across the six DNN regression models: (a) EfficientNetB0, (b) MobileNetV2, (c) NASNetMobile, (d) ResNet50, (e) DenseNet121, (f) InceptionV3. Each plot demonstrates the alignment between actual and predicted values, with the red dashed line representing perfect prediction.** Models with tighter clustering around the perfect prediction line, such as DenseNet121 and InceptionV3, exhibit higher prediction accuracy, whereas models like NASNetMobile show greater deviations.

**Table 5. Descriptive Statistics of Soil Moisture by Depth.**

|  | Deep | Middle | Top |
|---|---|---|---|
| count | 519 | 519 | 519 |
| mean | 65.33±15.01 | 41±11.74 | 21.26±8.29 |
| min | 23 | 5 | 0 |
| 25% | 55 | 29 | 16 |
| 50% | 70 | 44 | 22 |
| 75% | 77 | 51 | 27 |
| max | 93 | 64 | 42 |

observed distribution confirms that water supplied from the top infiltrates the soil and contributes significantly to moisture retention in the deep layer.

### 4.2.2. Regression models for depth-wise prediction.

Simple linear regression using soil moisture at the upper or middle depths exhibited only limited explanatory power for the deep layer, thereby underscoring the need for more advanced nonlinear approaches. Various nonlinear regression models were evaluated to predict soil moisture at different depths (Table 6, Fig 8, left). The two tree-based models (i.e., random forest regression and bagging decision tree regression) showed the best performance, with test $R^2$ of 0.906 and 0.903, respectively. The random forest model maintained a consistent gap between the training $R^2$ (0.942) and cross-validation $R^2$ (0.842), remaining within 10%, indicating strong generalization capability. Additionally, the standard deviation of the prediction error remained below 0.005 across different depths and moisture conditions, demonstrating stable predictive performance. The bagging decision tree model showed a 7% difference between training and cross-validation $R^2$, confirming its robustness and low risk of overfitting. These results suggest that ensemble learning approaches based on decision trees, such as random forest, effectively capture the complex nonlinear relationships inherent in soil moisture dynamics. In contrast, polynomial regression and KNN regression yielded lower cross-validation $R^2$ (0.762 and 0.816, respectively), indicating limited accuracy. This finding suggests that these models were less effective in modeling the nonlinear variability of soil moisture compared to tree-based methods.

Model hyperparameters were selected based on Occam's razor principle and the criterion that the difference between training and cross-validation $R^2$ remains within 10%. For polynomial regression, the polynomial degree varied from 2 to 5 to evaluate performance. Across all degrees, the test, training, and cross-validation $R^2$ values remained consistent at approximately 80.0%, 77.6%, and 76.2%, respectively. The performance gap between training and cross-validation was maintained at 1.8%, which is well below the 10% threshold. Therefore, based on Occam's razor, the second-degree polynomial was selected as the optimal hyperparameter due to its minimal computational cost. For SVR, regularization

**Table 6. Performance comparison of six machine learning regression models for soil moisture prediction. Reported $R^2$ and RMSE correspond to the best cross-validated hyperparameters. Random forest and bagging decision tree regression achieved the best predictive performance, followed by gradient boosting. Polynomial regression and KNN regression showed moderate performance, while SVR with an RBF kernel yielded the lowest. All models were evaluated on the same dataset containing multi-depth soil moisture measurements.**

|  | $R^2$ | RMSE | Optimal Hyperparameters |
|---|---|---|---|
| Polynomial Regression | 80.0 | 7.24 | Degree = 2 |
| Support Vector Regression | 76.7 | 9.75 | C = 10, RBF kernel |
| Bagging Decision Tree | 90.3 | 5.05 | n_estimators = 20 |
| Random Forest Regression | 90.6 | 4.97 | n_estimators = 50 |
| KNN Regression | 86.8 | 5.89 | n_neighbors = 3 |
| Gradient Boosting | 88.6 | 5.45 | n_estimators = 50 |

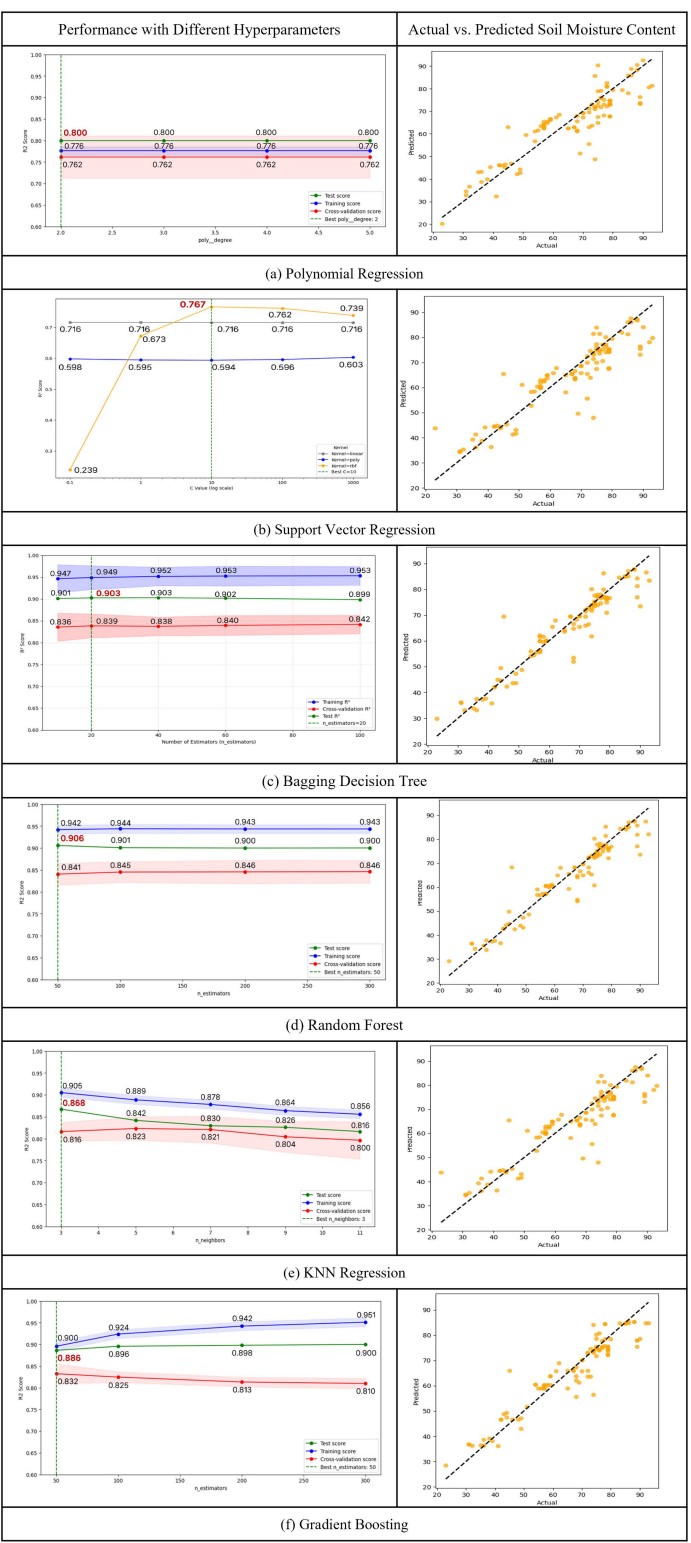

**Fig 8. Performance evaluation and comparison of six machine learning models for soil moisture prediction.** Left panels display R² scores across different hyperparameter values. Red shaded areas represent the standard deviation from k-fold cross-validation on the test set, and blue lines indicate training performance. Right panel is a scatter plot of actual vs. predicted soil moisture on the testing data using the optimized hyperparameter. The black dashed line indicates the 1:1 reference, and tighter clustering of points along this line reflects better predictive performance.

parameter C varied from 0.1 to 1000. When using the radial basis function (RBF) kernel, the model achieved the highest $R^2$ of 76.7% at C = 10. The RBF kernel outperformed both the linear and polynomial kernels, which yielded a lower $R^2$ of 71.6% and 59.4%, respectively. However, as C increased beyond 10, performance declined, with the $R^2$ dropping to 73.9% at C = 1000.

To optimize the hyperparameters for the tree-based models (i.e., bagging decision tree, random forest, and gradient boosting), the number of trees was adjusted and evaluated. For the bagging decision tree model, the number of estimators varied from 10 to 100. While the highest cross-validation $R^2$ of 84.2% was observed at 100 trees, the performance at 20 trees was only marginally lower at 83.9%, a difference of just 0.3%. Although the training–validation gap was slightly smaller at 100 trees, we selected 20 trees as the optimal hyperparameter considering the minimal performance difference, computational cost, and model complexity. For random forest, the number of trees was adjusted between 50 and 300. The highest cross-validation $R^2$ of 84.6% was achieved with 200 trees; however, the performance at 50 trees was nearly equivalent at 84.1%, with only a 0.5% difference. For the gradient boosting model, while the training $R^2$ increased from 90.0% to 95.1% as the number of trees increased, the cross-validation $R^2$ showed a declining trend (from 83.2% to 81.0%) indicating signs of overfitting. All three models demonstrated comparable or even better performance with a smaller number of trees. Therefore, based on Occam's razor and computational efficiency, the final number of estimators was set to 20 for the bagging decision tree model, and 50 for both random forest and gradient boosting.

For the KNN regression model, the number of neighbors varied from 3 to 11 for performance evaluation. The highest predictive performance was observed when the number of neighbors was set to 3, yielding a training $R^2$ of 90.5%, a cross-validation $R^2$ of 81.6%, and a test $R^2$ of 86.8 As the number of neighbors increased, all performance metrics gradually declined. Based on these results, 3 neighbors were selected as the optimal hyperparameter setting.

Fig 8 (right) illustrates the predictive performance of six nonlinear regression models (i.e., polynomial regression, SVR, decision tree regression, random forest regression, KNN regression, and gradient boosting) by comparing the actual and predicted soil moisture values. random forest regression exhibited the highest level of agreement between the predicted and actual values, indicating the strongest predictive performance. A majority of the data points were tightly clustered around the ideal prediction line, suggesting that the model could predict soil moisture with high accuracy and consistency across various soil conditions, while minimizing prediction errors. Similarly, the decision tree regression model demonstrated strong predictive accuracy, with most data points located near the prediction line. This finding indicates that the model effectively captured the nonlinear characteristics of the data. In contrast, the predicted values from the Polynomial Regression model were more widely scattered, reflecting a moderate level of agreement with the actual values. Several data points exhibited large prediction errors, suggesting that the model struggled to fully capture the nonlinear variability in the dataset. SVR showed the weakest predictive performance, with many data points deviating significantly from the ideal line. This result indicates that the model was unable to effectively learn the underlying data patterns and exhibited poor generalization in soil moisture prediction.

## 5. Discussion

### 5.1. Customized soil moisture management at plant level

To better understand the unique growth environments and moisture requirements of individual non-timber forest products, this study developed a non-invasive, customized soil moisture management model by integrating image-based machine learning with sensor data. The approach was designed to overcome limitations of conventional smart farming systems that rely on uniform control based on average environmental conditions. These systems often overlook the unique growth needs of individual plants. Although sensor-based systems have advanced environmental monitoring capabilities, they are typically applied in a generalized manner across entire cultivation areas, making it difficult to reflect plant-level variability. To address these limitations, this study proposes an alternative approach that enables customized management by predicting soil moisture using only surface images, thereby minimizing the need for dense sensor deployment. The study was

conducted at the seedling stage of wild-simulated ginseng, where soil remains visually accessible before being covered by foliage. While this stage-specific focus limits generalization across the entire growth cycle, it nonetheless underscores a practical contribution, as irrigation management is particularly critical during early development.

## 5.2. Comparison with previous approaches

The model was trained using RGB images of the soil surface paired with sensors at multiple depths that collected soil moisture values. This pairing allowed the model to learn the visual features associated with moisture distribution, enabling it to estimate soil moisture levels even in locations where sensors were not installed. This cross-referenced learning strategy bridges the gap between detailed sensing and scalable application, offering a practical solution for precise soil monitoring in resource-limited environments.

Previous sensor-based studies, such as Lloret et al. (2021), required dense sensor networks to capture subsurface moisture variability. In contrast, our image–sensor paired approach reduces sensor dependency by leveraging visual cues. Similarly, while recent remote-sensing efforts have mapped soil moisture at field or regional scales using machine learning with satellite or UAV imagery (Peng et al., 2024; Lamichhane et al., 2025) and UAV-based RGB sensing at local scales (Hernandez et al., 2025), our results demonstrate that RGB imagery alone can yield robust predictions at the plant scale. Deep learning models effectively extracted moisture-related features from soil textures, while ensemble-based regression models such as Random Forest captured nonlinear patterns in deeper soil layers. Notably, upper-layer moisture information emerged as a reliable predictor of subsurface moisture conditions. This finding suggests that surface imagery can, under certain conditions, serve as a proxy for deeper soil insights, offering a practical alternative in cases where intrusive sensing is infeasible. While deep-layer estimation is constrained by physical limitations, its potential value in sensor-limited environments warrants further exploration.

## 5.3. Practical contributions and agricultural implications

Beyond moisture estimation, the core contribution of this research lies in enabling plant-level management strategies based on non-destructive image data. Traditional cultivation practices often apply uniform treatments across crops, overlooking microenvironmental variation and individual physiological differences. Previous plant-based irrigation studies have emphasized the importance of individualized management (Jones, 2004), but practical methods have remained limited due to sensing costs and scalability issues. Our approach addresses this gap by allowing image-based assessment of plant-specific water status, supporting adaptive decisions tailored to each plant's developmental stage and soil environment.

Similar to recent imaging studies that evaluated leaf or canopy water status using hyperspectral and phenotyping platforms (Furbank & Tester, 2011; Ge et al., 2018), our framework demonstrates that even simple RGB imagery can provide actionable insights at the plant scale. Rather than merely improving irrigation efficiency, our approach promotes customized crop care, contributing to sustainable and resource-efficient cultivation. This aligns with the broader objectives of climate-smart agriculture (Lipper et al., 2014) by offering an accessible, sensor-free alternative for precision monitoring, particularly valuable in areas with limited infrastructure or water availability. Moreover, this system is especially suitable for managing high-value or sensitive crops, where minimal physical disturbance is essential. In future applications, the image-based approach shows potential for broader implementation. In this study, we primarily employed smartphone-based imaging to ensure accessibility for farmers, as smartphones are the most readily available and easy-to-use cameras in practice. This choice highlights the feasibility of applying our method for plant-level precision management in smallholder or high-value crop settings. However, direct scale-up to large-scale farming systems presents challenges, as smartphone-based imaging is not practical for monitoring extensive areas. To address this limitation, future implementations may require integration with UAV platforms, distributed camera networks, or hybrid approaches combining imaging with

sensor-based monitoring. Thus, while our current framework demonstrates clear value at the individual plant level, particularly for sensitive or high-value crops, its large-scale application will necessitate complementary technologies.

### 5.4. Limitations and future directions

Nevertheless, the study has certain limitations. We trained the model on data from a single crop, wild-simulated ginseng, under controlled laboratory conditions, which may affect generalizability. In addition, the current system relies exclusively on RGB images, which could limit sensitivity to subtle soil features. Future research should explore how to apply this method to diverse crop types and field environments and incorporate additional spectral data such as thermal or multispectral imagery to enhance prediction robustness. In addition, developing and testing indices that directly reflect changes in soil moisture could provide more sensitive predictors beyond raw spectral inputs. Furthermore, integrating this system with plant growth models could enhance its ability to support adaptive, plant-level crop management strategies throughout the entire growth cycle, enabling more context-aware decisions beyond irrigation alone.

At the same time, the reliance on smartphone-based imaging, while intentionally chosen to ensure accessibility for farmers, presents challenges when considering scalability. Smartphones are the most available and user-friendly tools for smallholders or high-value crop settings, but their use is not practical for monitoring large agricultural areas. Thus, while our framework demonstrates clear value at the individual plant level, its implementation in large-scale systems will likely require complementary technologies such as UAV platforms, distributed camera networks, or hybrid imaging–sensor solutions. Another constraint is the seedling-stage focus of our dataset. Although early-stage monitoring provides valuable insights when soil surfaces are visible, canopy closure at later growth phases limits direct soil observation. This necessitates further research of canopy-penetrating modalities, including multispectral or thermal imaging, and coupling with plant growth models to extend applicability throughout the crop cycle.

Although this study successfully developed depth-specific soil moisture models using sensor measurements collected at 15 cm, 10 cm, and 3 cm above the pot base, the surface-image–based component was applied only to the shallow layer. As a result, the present framework does not yet integrate surface imaging into an end-to-end system for predicting deeper soil moisture. Developing such an integrated approach remains a future goal, but would require corresponding ground-truth data for deeper layers as well as methods capable of addressing the substantial spatial heterogeneity of soil moisture conditions. Previous studies have similarly reported that machine learning based soil moisture models often exhibit reduced performance when applied across different locations or soil conditions unless multi-site calibration or domain adaptation techniques are employed [77–80]. Future extensions of the framework will therefore require multi-environment datasets and modeling strategies that explicitly account for spatial heterogeneity in soil moisture dynamics.

Soil moisture at transplantation was adjusted to approximately 40–60% of saturation to provide a uniform starting condition, and subsequent irrigation and natural drying generated substantial variation throughout the experiment. The dataset therefore included a wide spectrum of soil moisture levels, allowing the model to learn from heterogeneous conditions rather than being confined to a narrow range. Nevertheless, systematic evaluation under more extreme or fluctuating regimes, as encountered in field environments, represents an important avenue for future research to strengthen robustness and generalizability.

Resolution also constitutes a limitation, as training images were standardized to 224 × 224 pixels. While this choice ensured compatibility with convolutional neural network architectures and facilitated efficient model training, the trade-offs among image resolution, computational cost, and predictive robustness remain insufficiently explored. Future studies should systematically examine these trade-offs, as different resolutions may prove more effective depending on crop type, imaging conditions, and computational resources. Estimating soil moisture in deeper layers remains an open challenge. Future research may explore whether surface-based approaches can provide meaningful insights into subsurface dynamics. These limitations highlight promising directions for further research and emphasize that while the present framework

is suited for plant-level monitoring in controlled or high-value crop settings, its extension to large-scale agriculture will depend on methodological innovations and integration with complementary sensing technologies.

## 6. Conclusion

This study proposes a non-invasive soil moisture management model that integrates image-based machine learning with sensor data to support plant-specific management decisions. Among the evaluated deep learning models, DenseNet121 achieved the most accurate surface moisture prediction, while Random Forest Regression performed best for depth-wise estimation by capturing nonlinear moisture dynamics. The results demonstrate that surface image data can serve as a reliable proxy for subsurface soil conditions, reducing the need for extensive sensor deployment. This approach enables customized, plant-level crop management tailored to heterogeneous growing conditions, while minimizing physical disruption and operational costs. The system offers a scalable, cost-effective alternative to traditional uniform management strategies, particularly in the case of high-value or sensitive crops.

While the model showed strong performance, it was developed and tested under controlled conditions using a single crop. Future research should validate its applicability across diverse crop species, soil types, and field environments. Incorporating additional spectral data such as thermal or multispectral imagery may further improve model robustness and accuracy. Ultimately, this system lays the foundation for intelligent, adaptive crop management platforms, where AI not only predicts environmental conditions but also guides real-time, data-driven cultivation strategies across the entire crop lifecycle.

## Author contributions

**Conceptualization:** Yooan Kim, Sungyong Lee, Suhyun Lee, Kyo Suh.

**Formal analysis:** Yooan Kim.

**Methodology:** Yooan Kim, Suhyun Lee, Kyo Suh.

**Supervision:** Kyo Suh.

**Validation:** Yooan Kim, Suhyun Lee, Kyo Suh.

**Visualization:** Yooan Kim.

**Writing – original draft:** Yooan Kim, Kyo Suh.

**Writing – review & editing:** Yooan Kim, Taehyeong Kim, Sungyong Lee, Kyo Suh.

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
