## [Decision Letter · Decision Letter 0]

24 Jun 2025

Dear Dr. Suh,

Thank you for submitting your manuscript to PLOS ONE. After careful consideration, we feel that it has merit but does not fully meet PLOS ONE’s publication criteria as it currently stands. Therefore, we invite you to submit a revised version of the manuscript that addresses the points raised during the review process.

We look forward to receiving your revised manuscript.

Kind regards,

Babak Mohammadi

Academic Editor

PLOS ONE

Journal Requirements:

**Additional Editor Comments:**

Dear Authors,

The manuscript has been evaluated by three experts. Please provide point-to-point feedback, revisions, and/or responses to each comment of the reviewers. Besides, please consider the following comments during the revision and modify the manuscript accordingly, and provide feedback and modifications for each comment:

- Abstract, Results, Discussion, and Conclusion sections need more interpretation, discussion, and key findings in terms of Soil Moisture.

- Please investigate more literature review in the Introduction and then summarize their findings and make linkages between them and the research gaps you want to address in this study.

- Table 4: please provide statistical metrics for training and testing phases separately.

- Table 4: please also include Relative Root Mean Squared Error (RRMSE) as a new metric.

- Figure 2: please provide such figures for the training and testing phases separately.

- Figure 5 needs more explanation and discussion in the text.

Reviewers' comments:

Reviewer's Responses to Questions

**Comments to the Author**

1. Is the manuscript technically sound, and do the data support the conclusions?

Reviewer #1: Yes

Reviewer #2: Yes

Reviewer #3: Yes

2. Has the statistical analysis been performed appropriately and rigorously?

Reviewer #1: Yes

Reviewer #2: Yes

Reviewer #3: Yes

3. Have the authors made all data underlying the findings in their manuscript fully available?

Reviewer #1: Yes

Reviewer #2: Yes

Reviewer #3: No

4. Is the manuscript presented in an intelligible fashion and written in standard English?

Reviewer #1: Yes

Reviewer #2: Yes

Reviewer #3: Yes

Reviewer #1: This manuscript presents a very interesting approach involving image recognition and the use of AI to estimate soil moisture. I have a few minor comments to help improve the quality of the manuscript:

- Please add relevant references between lines 19 and 26.

- In lines 40–41, to which studies are the authors referring? Please clarify.

- Similarly, which studies are referenced in line 43? Please review the Introduction section to ensure all citations are properly included.

- The content between lines 72 and 82 appears to be background information. I suggest synthesizing this content and incorporating it into the Introduction.

- In lines 104–105, is the soil moisture reported on a gravimetric or volumetric basis? Please clarify this point.

- In Section 2.2, the authors describe the soil moisture monitoring system. Arduino-based soil moisture sensors are known for their limited accuracy (see for example https://doi.org/10.3390/s23052451), which can harm actual soil water conditions. Additionally, the manuscript mentions that only two points were used for sensor calibration. How can the authors ensure the reliability of the sensors using only a two-point calibration approach for Arduino-based sensors?

- In line 130, the authors mention a “100%” volumetric water content. Would this not refer to the degree of saturation of the soil sample rather than its volumetric water content? Please verify and ensure that the correct concepts and terminology are used consistently throughout the manuscript.

- The content between lines 239 and 246 also appears to be background information. Please remove it from Section 2.4.2 and integrate a synthesized version into the Introduction.

- In lines 424–426, could the reduced reliability of the soil moisture data have influenced the findings? Please consider and address this possibility.

- In lines 435–436, the authors seem to be referring again to the degree of saturation. If so, please correct the terminology.

- Currently, the Discussion section summarizes the main findings and highlights the study's limitations and contributions. However, a typical Discussion should also compare the results with findings from other studies. I strongly recommend revising this section to include comparisons with relevant recent literature.

Reviewer #2: • Maintain consistent use of the Oxford comma throughout lists.

• Replace passive voice were unnecessary, especially in methodology sections, with active constructions for clarity.

• Avoid stacking multiple clauses in one sentence; break them into two where necessary.

• Use consistent units of measurement (e.g., always clarify if % is gravimetric or volumetric moisture content).

• Use either American or British English consistently (e.g., “analyze” vs. “analyse”).

Reviewer #3: In this manuscript, the authors present a non-destructive image-based approach for estimating soil moisture. Top-view RGB images from a mobile camera were used as predictors for an adopted version of pre-trained deep learning architectures to predict the soil moisture content at the top layer. DenseNet121 was reported as the optimal model for soil moisture estimation in the top layer with high accuracy and computational efficiency. Moreover, non-linear regression models were optimized to allow for soil-moisture estimation at the bottom layer. The random forest and bagging decision tree performed the best in estimating the soil moisture at the deep layer from the measured soil moisture at the top and middle layers. Generally, the manuscript is well-written and structured in a logical order with well-justified choices. However, a few points need to be addressed before this work can be published.

**Major comments**

1) I recommend adding a timeseries figure including both actual and predicted soil moisture (of Figure 4 and Figure 5). This allows for further understanding of the capabilities and limitations of the image-based estimation. If this is not possible, please explain why not?!

2) What is the coefficient of determination between soil moisture at the upper and middle depths against the bottom layer? This check is important to understand the added value of the regression models used to predict the bottom layer.

3) The discussion of the manuscript can be further strengthened. Here are some points that are worth discussing:

- The study focused on the seedling stage, which raises the question of robustness and generalizability of your approach to the different growth stages. For example, a shadow will be introduced as the growth progresses, suggesting an updated model training, thus reducing its practicality.

- The approach applied looks infeasible for the intended purpose of crop management on a large scale. A key question would be how such an approach can be feasibly implemented for large-scale areas.

- Since the initial soil moisture was controlled (40-60%), how would changes in the initial soil moisture impact the performance of the AI model? Try to discuss this point and explain what is needed to quantitatively analyze this sensitivity.

- What was the rationale behind selecting 224x224 pixels? What is the implication of such a decision?

- The practicality does not fully convince me of your second part (i.e., soil moisture estimation at deep layers from the upper and middle measurements). In which cases can such a model be useful? The rationale behind this analysis requires strong arguments to accept it.

**Minor comments**

1) Lines 43-45: an example of references is missing.

2) Figure 1: Add dimensions to the cultivation system with more elaboration on its different components.

3) Figure 2: Add scale per subpanel.

4) Lines 171-172: add references to the pre-trained models.

5) Lines 167, 407: Typo in figure numbering

6) Table 3: caption is not complete. “is used to reduce…”

7) Lines 336-338: give different numbers for each question. Moreover, the equation of MSE is missing, and the second equation is for NRMSE, not RMSE. Make the necessary changes!

8) I found the heading and subheading titles are long. I suggest making shorter titles.

9) Table 4 should be split into Table 4 and Figure 4. Moreover, I suggest adding a legend that links the difference in the MAE values and the bubble size.

10) Table 6: The ranking of the models is not correct in the caption. Change it to “random forest and bagging decision tree regression … followed by gradient boosting”.

11) Figure 5: Add more information about the right panels in the caption. For example, they are based on the testing data and the optimized hyperparameter indicated by the green vertical dashed line, or so!

12) Type in line 550, change “Furesearch” to “Future research”.

13) Line 552: future work should explore other predictors beyond the measured spectrum (i.e., indices that can indicate changes in soil moisture).

Awad M. Ali

Hydrology and Environmental Hydraulics Group,

Wageningen University

**Do you want your identity to be public for this peer review?** For information about this choice, including consent withdrawal, please see our Privacy Policy

Reviewer #1: **Yes:** Luis Eduardo Bertotto

Reviewer #2: **Yes:** Dr. Iftikhar Ahmed

Reviewer #3: **Yes:** Awad Mohammed Ali

---

## [Author Response · Author response to Decision Letter 1]

1 Sep 2025

Manuscript Reference: PONE-D-25-27146

Title: Image-Based Machine Learning Models for Customized Soil Moisture Management

Journal: PLOS One

Reviewer #1

We would like to say thank you for your valuable comments to improve our paper. We have addressed the comments as explained below.

[1] Please add relevant references between lines 19 and 26.

Thank you for your comment. As suggested, we have added relevant references between lines 19 and 26 to strengthen the background and support the statements. The revised text now includes recent studies on smart farming and water resource management (Parra-López et al., 2025; Neophytides et al., 2024; Ingrao et al., 2023; Al-Kaisi & Broner, 2014; Kelly et al., 2025; Bounajra et al., 2024).

In manuscript:

(Line 19-26)

In the evolution of smart farming, water resource management technology has gained significant attention (Parra-López et al., 2025; Neophytides et al., 2024). As climate change intensifies water scarcity, efficient water resource management in agriculture has become an increasingly critical challenge (Ingrao et al., 2023). Moisture management in crop cultivation is a key factor in determining productivity and quality (Al-Kaisi & Broner, 2014). For example, inadequate moisture management can cause direct economic losses through reduced plant growth, increased pest and disease occurrence, and decreased yields (Ingrao et al., 2023). Precise moisture management requires accurate understanding crop water requirements based on growth stages and environmental conditions, and then providing the appropriate amount of water for each plant at the right time (Kelly et al., 2025; Al-Kaisi &Broner, 2014; Bounajra et al., 2024).

[2] Similarly, which studies are referenced in line 43? Please review the Introduction section to ensure all citations are properly included.

We have revised the citation for line 43 to ensure accuracy, and the sentence now cites European Parliamentary Research Service (2023), Akkem et al. (2023), and Soussi et al. (2023) as the relevant studies. In addition, as suggested, we reviewed the entire Introduction section to ensure the section is supported by relevant literature, and we have added citations where necessary.

In manuscript:

(Line 43)

Most artificial intelligence research related to smart farms incorporating various digital technologies has primarily focused on uniform management at the field or greenhouse level. Limited studies exist on customized cultivation management systems that consider the growth status of individual crops (European Parliamentary Research Service, 2023; Akkem et al., 2023; Soussi et al., 2023).

[3] The content between lines 72 and 82 appears to be background information. I suggest synthesizing this content and incorporating it into the Introduction.

Thank you for your comment. We carefully reviewed the content between lines 72 and 82 and agree that it provides important background information on the target crop. While your suggestion was to incorporate this content into the Introduction, we determined that it is more closely related to the description of experimental materials and design. Therefore, in revising the manuscript, we reorganized Section 2.1 Experimental Setup for Customized Growth Management to improve clarity and logical flow, creating two subsections: 2.1.1 Target Crop: Wild-simulated Ginseng and 2.1.2 Experiment Setup. The content originally found between lines 72 and 82 has been moved to the Target Crop subsection, where it directly supports the experimental design.

Below is the revised text for 2.1.1 Target Crop: Wild-simulated Ginseng, extracted from the reorganized Section 2.1 Experimental Setup for Customized Growth Management. Only the relevant subsection is presented here for clarity.

In manuscript:

(Line 79-92)

2.1.1. Target Crop: Wild-simulated Ginseng

High value-added agricultural and forest products command premium prices relative to their yield, highlighting the need for customized management to maximize productivity. The market price per gram of wild-simulated ginseng (WSG) exceeds that of cultivated ginseng by more than 20-fold, highlighting its prominence as a high-value forest product (Han et al., 2017; Jeong et al., 2019). Most WSG is marketed after more than seven years of cultivation. The quality and price are determined by various factors such as region, shape, weight, and other attributes that affect market value (Moon et al., 2019; Park et al., 2013; Shaoqing et al., 2013). Since the survival rate of WSG decreases dramatically from the fourth year onward, effective management strategies are required to improve survival rates and produce high-quality ginseng (Korea Forest Service, 2019; Woo, 2016). To establish legal standards for WSG and clearly distinguish it from cultivated ginseng, we investigated changes in WSG characteristics after a short post-treatment period following pre-germination transplantation. Accordingly, five-year-old WSG cultivated in Pyeongchang, Gangwon Province, was selected and transplanted into experimental soil as the target forest product for this study.

[4] In lines 104–105, is the soil moisture reported on a gravimetric or volumetric basis? Please clarify this point.

Thank you for your comment. The soil moisture content (50 ± 10%) reported in lines 104–105 was expressed as a percentage of the fully saturated condition of the soil, based on two-point calibration (0% for oven-dry soil and 100% for saturation). While the sensor output was calibrated using volumetric water content readings from a pre-calibrated device, the reported value represents the relative saturation level rather than an absolute volumetric water content. We have clarified this distinction in the revised manuscript.

In manuscript:

(Line 112-116)

Following transplantation, the soil moisture content was adjusted to 50 ± 10% of saturation, based on two-point calibration (oven-dry and fully saturated conditions), to provide a suitable starting environment. In this study, soil moisture content refers to the relative saturation of the soil, expressed as a percentage with 0% for oven-dry soil and 100% for saturation. Hereafter, this is referred to simply as soil moisture content.

[5] In Section 2.2, the authors describe the soil moisture monitoring system. Arduino-based soil moisture sensors are known for their limited accuracy (see for example https://doi.org/10.3390/s23052451), which can harm actual soil water conditions. Additionally, the manuscript mentions that only two points were used for sensor calibration. How can the authors ensure the reliability of the sensors using only a two-point calibration approach for Arduino-based sensors?

Thank you for your comment and for highlighting the potential limitations of Arduino-based soil moisture sensors. We acknowledge that such sensors, in general, have lower absolute accuracy compared to high-grade instruments. In our study, to minimize this limitation, we individually calibrated each Arduino-based sensor against a pre-calibrated professional soil moisture meter prior to the experiment. The professional device had undergone factory calibration, ensuring a reliable reference for the calibration process.

Although a high-grade instrument was available, we opted to use Arduino-based sensors because our experimental design required simultaneous real-time monitoring of soil moisture across multiple pots. This was not feasible with a single high-grade device. Among the various types of Arduino-compatible sensors, we selected capacitive sensors, which are known to be more resistant to corrosion than resistive types. Furthermore, the experiment was conducted over a relatively short period, reducing the risk of sensor degradation. These measures, combined with controlled environmental conditions, allowed us to maintain sufficient reliability for our experimental objectives while addressing the known limitations of Arduino based sensors.

[6] In line 130, the authors mention a “100%” volumetric water content. Would this not refer to the degree of saturation of the soil sample rather than its volumetric water content? Please verify and ensure that the correct concepts and terminology are used consistently throughout the manuscript.

Thank you for your comment. As clarified in our response to Comment [4] and reflected in the revised Section 2.1, we defined “soil moisture content” based on a two-point calibration (0% for oven-dry soil and 100% for saturation). Therefore, the value referred to as “100%” in the original text indicates full saturation rather than volumetric water content in the strict sense. To maintain consistency, we have revised the manuscript to use the term “soil moisture content” throughout, which hereafter denotes relative saturation as defined in Section 2.1.

In manuscript:

(Line 140-141)

Based on two-point calibration, the average sensor output was 542 ± 54 at 0% (oven-dry) and 344 ± 50 at 100% (saturation).

[7] The content between lines 239 and 246 also appears to be background information. Please remove it from Section 2.4.2 and integrate a synthesized version into the Introduction.

Thank you for your helpful comment. In accordance with your suggestion, we have removed the background description from Section lines 239–246 in the original manuscript and integrated the relevant content into the Introduction to improve the manuscript.

In manuscript:

(Line 30-33)

Since each soil layer plays a distinct role in crop growth, understanding soil moisture variation by depth is also essential (He et al., 2021). The upper layer reflects rapid irrigation and evaporation changes (USDA, 1991), the middle layer is the primary zone for root activity, and the bottom layer functions as a reservoir that buffers excess moisture (Bai et al., 2025; He et al., 2025).

[8] In lines 424–426, could the reduced reliability of the soil moisture data have influenced the findings? Please consider and address this possibility.

Thank you for the comment. In the section referenced (lines 424–426), the increased prediction uncertainty at high-moisture conditions arises primarily from image-domain factors rather than degraded label reliability. Near saturation, the soil surface becomes darker and more homogeneous, pools of water introduce specular reflections, and the peatmoss/perlite mixture reduces micro-texture contrast, collectively weakening the visual cues available to the model. By design, ground-truth moisture labels were obtained from short-term, individually calibrated sensors and data were collected before canopy occlusion, which limited label noise. We have added a note to clarify that the observed uncertainty is mainly due to diminished discriminability in surface imagery under high moisture, while label reliability was managed through calibration and controlled acquisition.

In manuscript:

(Line 431-436)

Across all models, there was a general trend of increased prediction uncertainty under high-moisture conditions, which is likely due to the subtle changes in the soil texture that complicate accurate prediction. The findings highlight that an effective feature extraction strategy plays a more critical role in predictive performance than model complexity alone. Notably, the higher uncertainty under near-saturation is attributable to reduced visual discriminability of the soil surface (darkening, specular reflections, and homogenization of the peatmoss/perlite mixture), which limits image-based cues.

[9] In lines 435–436, the authors seem to be referring again to the degree of saturation. If so, please correct the terminology.

We appreciate the reviewer’s comment regarding the terminology. To address this, we clarified in the Methods section that soil moisture content in this study is defined as relative saturation (%) based on two-point calibration (oven-dry and fully saturated). We also revised Section 4.2.1 to explicitly state that the reported values represent relative saturation.

In manuscript:

(Line 444-445)

The bottom layer showed the highest mean soil moisture content, measured as relative saturation, with a mean value of 65.33% and a standard deviation of 15.01%.

[10] Currently, the Discussion section summarizes the main findings and highlights the study's limitations and contributions. However, a typical Discussion should also compare the results with findings from other studies. I strongly recommend revising this section to include comparisons with relevant recent literature.

We thank you for this important suggestion. In the revised manuscript, we have substantially expanded the Discussion to include explicit comparisons with recent literature. Specifically, we now contrast our image–sensor paired approach with prior sensor-based methods that required dense sensor networks for subsurface monitoring (Lloret et al., 2021). We also compare our plant-scale results with recent remote sensing studies that used satellite or UAV imagery for soil moisture estimation at larger scales (Peng et al., 2024; Lamichhane et al., 2025; Hernandez et al., 2025). Furthermore, we discuss similarities with imaging studies that employed hyperspectral and phenotyping platforms for assessing canopy water status (Furbank & Tester, 2011; Ge et al., 2018). By situating our findings within these recent advances, we highlight the novelty of our approach namely, demonstrating that simple RGB imagery can enable robust, plant-level moisture estimation with reduced sensor dependency. These revisions strengthen the Discussion by clarifying how our study complements and extends existing research.

Reviewer #2

We would like to say thank you for your valuable comments to improve our paper. We have addressed the comments as explained below.

[1] Maintain consistent use of the Oxford comma throughout lists.

Thank you for your observation. In the revised manuscript, we carefully reviewed all lists and ensured the consistent application of the Oxford comma. For example, the list of models in the Results section was revised to read “ResNet50, EfficientNetB0, MobileNetV2, InceptionV3, DenseNet121, and NASNetMobile.” This consistent usage enhances readability and maintains stylistic uniformity throughout the manuscript.

[2] Replace passive voice were unnecessary, especially in methodology sections, with active constructions for clarity.

We appreciate your suggestion. We revised the manuscript to reduce unnecessary passive voice, particularly in the Methodology and Results sections. Whenever possible, sentences were rewritten in active voice to improve clarity and readability. For example, the previous phrasing “The number of trees was adjusted and evaluated” was revised to “We adjusted and evaluated the number of trees.” These revisions highlight the authors’ role in the research process and improve narrative flow.

[3] Avoid stacking multiple clauses in one sentence; break them into two where necessary.

We agree with your comment. Throughout the manuscript, sentences that contained multiple clauses were revised and divided into two or more concise statements to enhance readability. For instance, in the Results section, the sentence “As C increased beyond 10, performance declined, with the R² dropping to 73.9% at C = 1000” was revised into “As C increased beyond 10, performance declined. The R² dropped to 73.9% at C = 1000.” These changes improve clarity and ensure that the manuscript is easier to follow for readers.

[4] Use consistent units of measurement (e.g., always clarify if % is gravimetric or volumetric moisture content).

Thank you for raising this important point. We have revised the manuscript to clarify that soil moisture content is expressed as relative saturation, where 0 percent corresponds to oven-dry soil and 100 percent corresponds to fully saturated soil. To avoid ambiguity, we now consistently use this definition throughout the text and explicitly state that the values refer to relative saturation rather than gravimetric or volumetric content.

[5] Use either American or British English consistently (e.g., “analyze” vs. “analyse”).

Thank you for pointing this out. The manuscript has been carefully revised to ensure consistent use of American English spelling and style throughout the text.

Reviewer #3

We would like to say thank you for your valuable comments to improv

---

## [Decision Letter · Decision Letter 1]

29 Sep 2025

Dear Dr. Suh,

Thank you for submitting your manuscript to PLOS ONE. After careful consideration, we feel that it has merit but does not fully meet PLOS ONE’s publication criteria as it currently stands. Therefore, we invite you to submit a revised version of the manuscript that addresses the points raised during the review process.

We look forward to receiving your revised manuscript.

Kind regards,

Julfikar Haider

Academic Editor

PLOS ONE

Journal Requirements:

**Additional Editor Comments:**

Please find few comments to address

Reviewers' comments:

Reviewer's Responses to Questions

**Comments to the Author**

Reviewer #1: All comments have been addressed

Reviewer #3: (No Response)

2. Is the manuscript technically sound, and do the data support the conclusions?

Reviewer #1: Yes

Reviewer #3: Yes

3. Has the statistical analysis been performed appropriately and rigorously?

Reviewer #1: Yes

Reviewer #3: Yes

4. Have the authors made all data underlying the findings in their manuscript fully available?

Reviewer #1: Yes

Reviewer #3: Yes

5. Is the manuscript presented in an intelligible fashion and written in standard English?

Reviewer #1: Yes

Reviewer #3: Yes

Reviewer #1: (No Response)

Reviewer #3: I would like to thank the authors for their genuine consideration of my first review. However, few points remain to finalize the reviewing from my side.

- I noticed that my minor comment about Figure 2 was missed. Please add scale per subpanel.

- For point [2], I suggest adding the R2 values of the linear models for better clarification.

- For point [3], I believe my final suggestion was not fully addressed in the revised manuscript. Training a ML model to predict the deeper layer from the surface layer requires training hence information about the deeper soil moisture (from sensors?). Given that soil moisture can be largely heterogeneous, to what extent one can transfer the trained model to other locations. Please add some insights from literature to properly address my concern.

**Do you want your identity to be public for this peer review?** For information about this choice, including consent withdrawal, please see our Privacy Policy

Reviewer #1: **Yes:** Luis Eduardo Bertotto

Reviewer #3: **Yes:** Awad Mohammed Ali

---

## [Author Response · Author response to Decision Letter 2]

23 Nov 2025

We would like to say thank you for your valuable comments to improve our paper. We have addressed the comments as explained below.

[1] The manuscript does not provide sufficient detail on soil physical properties (e.g., texture, bulk density, field capacity), which are critical to understanding water retention behavior and image-based interpretation.

We appreciate your comment regarding the lack of detail on soil. In this study, all plants were grown in a commercially standardized peat moss and perlite mixture (6:4 ratio), which was intentionally kept constant across experimental units because the objective was to examine moisture stress responses and image-based predictions within a uniform substrate rather than to compare different soil types. For this reason, soil physical properties were not treated as independent experimental variables. However, we fully agree that the general physical characteristics of the substrate are important for understanding water retention behavior and sensor interpretation. To address this concern, we have revised the manuscript to include a description of the typical physical properties of peat–perlite mixtures based on manufacturer information and relevant literature (high porosity, low bulk density, and moderate-to-high water-holding capacity). We appreciate your helpful suggestion, which has strengthened the clarity and contextualization of the experimental setup.

In manuscript:

(Line 97-106) WSG seedlings were grown under controlled greenhouse conditions following transplantation. The image-based analysis in this study focused on the early growth period after transplantation, a stage during which wild-simulated ginseng is highly sensitive to moisture conditions. Moisture stress during this period has a direct influence on seedling survival and early physiological responses, including changes in saponin composition[53]. To ensure consistent rooting conditions across plants, we used a standardized peat moss and perlite substrate in a 6:4 ratio, which is commonly adopted for ginseng seedling cultivation[54, 55]. This substrate composition is known to provide high porosity, low bulk density, and a favorable balance between aeration and water-holding capacity. Such physical characteristics create a uniform and supportive rooting environment, which is particularly important during the moisture-sensitive early growth stage of WSG

[2] The sensor data appears to be in arbitrary units (0–1023) converted via calibration, yet final reported moisture predictions are in percentages.

Thank you for raising this point regarding the conversion of raw sensor outputs into soil moisture values. The capacitive sensors used in this study generate electrical signals in the 01023 range, which must be converted before meaningful interpretation is possible. To address this concern, we have clarified the calibration procedure in the revised manuscript. Specifically, we now explain that soil moisture (%) values were derived using a standard two-point calibration approach based on oven-dry (0%) and fully saturated (100%) reference states. This method converts the raw electrical signals into normalized moisture estimates rather than direct volumetric water content. We have updated the Methods section to explicitly describe this conversion process and to provide the corresponding calibration values. We believe these revisions improve transparency and strengthen the methodological clarity of the study. We sincerely appreciate your suggestion.

In manuscript:

(Line 130-142) Soil moisture status was measured using capacitive moisture sensors integrated into the Arduino-based moisture management system. Each sensor was installed individually for each WSG plant to enable customized monitoring and control of growth conditions. To avoid signal distortion, which is commonly caused by corrosion in resistive sensors, we used HW-390 capacitive soil moisture sensors. The capacitive soil moisture sensors used in this study detect changes in dielectric permittivity associated with varying soil moisture levels based on frequency domain reflectometry (FDR). These readings are converted into electrical signals ranging from 0 to 1023. Higher values correspond to lower moisture conditions. To convert the raw sensor outputs into normalized moisture (%) values, we applied a two-point calibration procedure using oven-dry (0%) and fully saturated (100%) reference states. The average sensor readings were 542 ± 54 at 0% and 344 ± 50 at 100%. In this study, the moisture (%) values represent a normalized index derived from the calibrated sensor signals rather than direct volumetric or gravimetric water content.

For the soil-depth analysis experiment, we installed three moisture sensors per plant at distinct heights above the pot base: 15 cm (top layer), 10 cm (middle layer), and 3 cm (deep layer). These layer designations were adopted to align with conventional soil profile terminology and to provide clearer interpretation of the soil structure within the pot.

[3] The reported moisture levels (e.g., >60%) seem high for peat-perlite mixes, unless the value is gravimetric or saturated conditions were used. Confirm the unit (volumetric vs gravimetric) and measurement technique used for "moisture %" throughout the manuscript.

We appreciate insightful comment regarding the interpretation of the moisture (%) values. We would like to clarify that the moisture percentages reported in this study do not represent direct volumetric or gravimetric water content. Instead, these values correspond to a normalized index derived from the calibrated electrical outputs of the capacitive sensors. The raw sensor signals (0–1023) were converted into normalized moisture (%) values using a two-point calibration procedure based on oven-dry (0%) and fully saturated (100%) reference conditions. As a result, values such as >60% indicate relative moisture levels with respect to saturation rather than absolute volumetric water content. To avoid any ambiguity, we have revised the Methods section to explicitly state that moisture (%) is presented as a normalized index rather than a physical water content measure. We thank you for this helpful suggestion, which has improved the clarity of the manuscript.

In manuscript:

(Line 135-137) In this study, the moisture (%) values represent a normalized index derived from the calibrated sensor signals rather than direct volumetric or gravimetric water content.

[4] Terms like “bottom layer = 3 cm from base” may confuse readers, as this is typically referred to as "deep layer." Consider using conventional soil profile terminology (surface, subsoil, etc.).

We sincerely appreciate your comment regarding the terminology used to describe the soil layers. As noted, the expression “bottom layer = 3 cm from base” may lead to confusion and does not align well with conventional soil profile terminology. In response, we have revised the manuscript to use clearer and more standard terms. Specifically, the sensors positioned 15 cm, 10 cm, and 3 cm above the pot base are now described as the top, middle, and deep soil layers, respectively. We believe this terminology provides a more intuitive understanding of the soil structure and improves the clarity of the experimental description. In addition, we have carefully reviewed the entire manuscript and updated all related terminology to ensure consistency throughout the text. We sincerely thank you for this helpful suggestion.

[5] Include soil moisture distribution profiles over time (e.g., after irrigation) to visualize how moisture moves through layers. A confusion matrix or regression scatterplots with residuals may further support the model’s predictive capability.

We sincerely appreciate your suggestions regarding the visualization of soil moisture dynamics and model performance. We would like to clarify that the present study did not track soil moisture redistribution over time after irrigation. Our experiment matched the soil moisture measured by sensors with the images captured at the same moment to train the models for estimating individual soil moisture conditions. Because continuous time-series data were not collected, we are unable to generate soil moisture distribution profiles over time. Nonetheless, we find your suggestion highly valuable and agree that incorporating temporal moisture redistribution analysis would be a meaningful direction for future advanced research.

Regarding your recommendation to include a confusion matrix or regression visualizations, we interpreted the suggestion for a confusion matrix as an interest in improving clarity in model evaluation. However, since the present study does not use a classification model, tools that are specifically designed for classification such as confusion matrices cannot be applied within our modeling framework. As for regression-based visualizations, scatterplots and residual distributions have already been provided in Figure 5 and Figure 6, which illustrate the predictive performance and residual patterns of the models. We appreciate your feedback and believe that the suggestions have helped strengthen the clarity of the manuscript.

[6] Some references are cited as "2024/2025" which may still be in press or unavailable. Ensure all references are accessible and published.

We appreciate your comment regarding the references listed as 2024/2025. We have thoroughly reviewed the reference list and corrected all instances of 2024/2025, ensuring that only accessible and fully published sources are cited in the revised manuscript. Thank you for pointing this out.

Reviewer [3]

We would like to say thank you for your valuable comments to improve our paper. We have addressed the comments as explained below.

[1] I noticed that my minor comment about Figure 2 was missed. Please add scale per subpanel.

We appreciate helpful suggestion regarding the inclusion of scale bars in Figure 2. Although the images were captured at a fixed camera distance, no physical reference object or measured scale was included during image acquisition. Under these conditions, it is not possible to derive an accurate physical scale, and adding an arbitrary scale bar could mislead readers by implying a level of spatial or dimensional accuracy that the images do not provide. Figure 2 is intended solely to offer a qualitative illustration of visual differences across moisture levels, rather than to support spatial or size-dependent interpretation. To avoid potential misunderstanding while maintaining clarity, we have added a note to the figure caption indicating that the images are not to scale.

In manuscript:

Figure 2 Sample soil surface images at different moisture levels. These images are provided solely for qualitative illustration, and no spatial or physical scale is implied (not to scale).

[2] I suggest adding the R2 values of the linear models for better clarification.

We sincerely appreciate your thoughtful comment regarding the inclusion of R² values. We would like to clarify that all R² values associated with the models used in this study are already fully reported in Table 4 and Table 6 of the revised manuscript. These values reflect the best-performing models obtained. However, in the event that we have misunderstood your intended meaning or if you were referring to a different form or context of R² beyond what has already been provided, we would be more than willing to incorporate any additional R² values or clarifications as necessary. We appreciate your careful evaluation and remain grateful for the feedback.

[3] Training a ML model to predict the deeper layer from the surface layer requires training hence information about the deeper soil moisture (from sensors?).

We sincerely appreciate the comment regarding the relationship between surface imaging and deeper-layer soil moisture estimation. To clarify the scope of the current study, the deeper-layer analysis was conducted using actual sensor measurements collected at 15 cm, 10 cm, and 3 cm above the pot base, and these data were used to train and evaluate the depth-specific prediction models. Therefore, the deeper-layer models were developed with the necessary information. At the same time, the present work does not implement an end-to-end system that predicts deeper soil moisture directly from surface images. The image-based component was designed only for surface-layer estimation, and integrating surface imagery with deeper-layer prediction remains a future research goal. We agree with the reviewer that achieving such integration would require careful consideration of soil heterogeneity and the challenges of transferring models across different conditions. We appreciate your insight, and we have revised the manuscript to better clarify the scope of the current framework and the implications for future development.

In manuscript:

(Line 570-579) Although this study successfully developed depth-specific soil moisture models using sensor measurements collected at 15 cm, 10 cm, and 3 cm above the pot base, the surface-image–based component was applied only to the shallow layer. As a result, the present framework does not yet integrate surface imaging into an end-to-end system for predicting deeper soil moisture. Developing such an integrated approach remains a future goal but would require corresponding ground-truth data for deeper layers as well as methods capable of addressing the substantial spatial heterogeneity of soil moisture conditions. Previous studies have similarly reported that machine learning based soil moisture models often exhibit reduced performance when applied across different locations or soil conditions unless multi-site calibration or domain adaptation techniques are employed[77-80]. Future extensions of the framework will therefore require multi-environment datasets and modeling strategies that explicitly account for spatial heterogeneity in soil moisture dynamics.

---

## [Decision Letter · Decision Letter 2]

14 Jan 2026

Image-Based Machine Learning Models for Customized Soil Moisture Management

PONE-D-25-27146R2

Dear Dr. Suh,

We’re pleased to inform you that your manuscript has been judged scientifically suitable for publication and will be formally accepted for publication once it meets all outstanding technical requirements.

Kind regards,

Gobinath Ravindran

Academic Editor

PLOS One

Additional Editor Comments (optional):

Reviewers' comments:

Reviewer's Responses to Questions

**Comments to the Author**

Reviewer #1: All comments have been addressed

Reviewer #3: All comments have been addressed

2. Is the manuscript technically sound, and do the data support the conclusions?

Reviewer #1: Yes

Reviewer #3: Yes

3. Has the statistical analysis been performed appropriately and rigorously?

Reviewer #1: Yes

Reviewer #3: Yes

4. Have the authors made all data underlying the findings in their manuscript fully available?

Reviewer #1: Yes

Reviewer #3: No

5. Is the manuscript presented in an intelligible fashion and written in standard English?

Reviewer #1: Yes

Reviewer #3: Yes

Reviewer #1: I have no further comments to the manuscript, therefore I recommend acceptance of the article in the current version.

Reviewer #3: I am satisfied with your response to my previous comments. I believe the manuscript is ready to be published. Well done!

**Do you want your identity to be public for this peer review?** For information about this choice, including consent withdrawal, please see our Privacy Policy

Reviewer #1: No

Reviewer #3: **Yes:** Awad M. Ali

---

## [Editor Report · Acceptance letter]

PONE-D-25-27146R2

PLOS One

Dear Dr. Suh,

I'm pleased to inform you that your manuscript has been deemed suitable for publication in PLOS One. Congratulations! Your manuscript is now being handed over to our production team.

Kind regards,

on behalf of

Dr. Gobinath Ravindran

Academic Editor

PLOS One